# Learning Equivariant Non-Local Electron Density Functionals

**Nicholas Gao**\*, **Eike Eberhard**\*, **Stephan Günnemann**
{n.gao,e.eberhard,s.guennemann}@tum.de
Department of Computer Science & Munich Data Science Institute
Technical University of Munich

## Abstract

The accuracy of density functional theory hinges on the approximation of *non-local* contributions to the exchange-correlation (XC) functional. To date, machine-learned and human-designed approximations suffer from insufficient accuracy, limited scalability, or dependence on costly reference data. To address these issues, we introduce Equivariant Graph Exchange Correlation (EG-XC), a novel non-local XC functional based on equivariant graph neural networks (GNNs). Where previous works relied on semi-local functionals or fixed-size descriptors of the density, we compress the electron density into an SO(3)-equivariant nuclei-centered point cloud for efficient non-local atomic-range interactions. By applying an equivariant GNN on this point cloud, we capture molecular-range interactions in a scalable and accurate manner. To train EG-XC, we differentiate through a self-consistent field solver requiring only energy targets. In our empirical evaluation, we find EG-XC to accurately reconstruct 'gold-standard' CCSD(T) energies on MD17. On out-of-distribution conformations of 3BPA, EG-XC reduces the relative MAE by $35\%$ to $50\%$. Remarkably, EG-XC excels in data efficiency and molecular size extrapolation on QM9, matching force fields trained on 5 times more and larger molecules. On identical training sets, EG-XC yields on average $51\%$ lower MAEs.

## 1 Introduction

Kohn-Sham Density Functional Theory (KS-DFT) is the backbone of computational material and drug discovery (Jones, 2015). It is a quantum mechanical method to approximate the ground state energy of an $N_{\text{el}}$-electron system by finding the electron density $\rho \in \mathcal{D}_{N_{\text{el}}} = \left\{\rho : \mathbb{R}^3 \to \mathbb{R}_+ | \int_{\mathbb{R}^3} \rho(r) dr = N_{\text{el}}\right\}$ that minimizes the energy functional $E : \mathcal{D}_{N_{\text{el}}} \to \mathbb{R}$. This functional maps electron densities to energies and is composed of

$$E[\rho] = T[\rho] + V_{\text{ext}}[\rho] + V_{\text{H}}[\rho] + E_{\text{XC}}[\rho] \tag{1}$$

$T : \mathcal{D}_{N_{\text{el}}} \to \mathbb{R}$ is the kinetic energy functional, $V_{\text{ext}} : \mathcal{D}_{N_{\text{el}}} \to \mathbb{R}$ the external potential due to positively charged nuclei, $V_{\text{H}} : \mathcal{D}_{N_{\text{el}}} \to \mathbb{R}_+$ the Coulomb energy between electrons, and, finally, $E_{\text{XC}} : \mathcal{D}_{N_{\text{el}}} \to \mathbb{R}_-$ the *exchange-correlation* (XC) functional (Cramer, 2004). While $T, V_{\text{ext}}$ and $V_{\text{H}}$ permit analytical computations, the exact form of $E_{\text{XC}}$ remains unknown, and its approximation frequently dominates DFT's error (Kim et al., 2013).

Machine learning has emerged as a promising data-driven approach to approximate $E_{\text{XC}}$ (Kulik et al., 2022; Zhang et al., 2023). Many such ML functionals adopt the classical approach and define $E_{\text{XC}}$ as the integral of a learnable XC energy density $\epsilon_{\text{XC}} : \mathbb{R}^{d_{\text{mGGA}}} \to \mathbb{R}$ (Perdew, 2001):

$$E_{\text{XC}}[\rho] = \int_{\mathbb{R}^3} \rho(r)\epsilon_{\text{XC}}\left(\boldsymbol{g}(r)\right) dr \tag{2}$$

where $\boldsymbol{g} : \mathbb{R}^3 \to \mathbb{R}^d$ are properties of the electron density $\rho$ (Dick & Fernandez-Serra, 2021; Nagai et al., 2022). If $\boldsymbol{g}$ only depends on density quantities from its infinitesimal neighborhood, e.g., $\rho(r), \nabla\rho(r), \ldots$, the resulting functionals are called *semi-local*. While this integrates well into existing quantum chemistry code, non-local interactions like Van der Waals forces exceed the functional

---

\*Equal contribution; corresponding authors.

class (Kaplan et al., 2023). In contrast, *non-local* functionals can capture such interactions by depending on multiple points in space simultaneously, e.g., $E_{\text{XC}}[\rho] = \iint_{\mathbb{R}^3} \rho(r)\rho(r')\epsilon_{\text{XC}}(\boldsymbol{g}(r), \boldsymbol{g}(r'))drdr'$. Non-locality can be further separated into *atomic-range* interactions, which depend on the electron density around the nucleus, and *molecular-range* interactions, which depend on the electron density within typical molecular length scales. But, existing non-local functionals either scale poorly computationally (Zhou et al., 2019) or require costly reference data (Margraf & Reuter, 2021; Bystrom & Kozinsky, 2022). The critical challenge in XC functionals lies in efficiently captureing non-local molecular-range interactions at scale.

To this end, we propose to leverage equivariant graph neural networks (GNNs) to learn non-local XC functionals through our Equivariant Graph Exchange Correlation (EG-XC)[1]. To enable computationally efficient non-local interactions, we propose two key innovations: (1) We compress the electron density to an SO(3)-equivariant atomic-range point cloud representation by convolving the electron density with an SO(3)-equivariant kernel at the nuclear positions. (2) We use SO(3)-equivariant GNNs on this point cloud representation to efficiently capture molecular-range information. Finally, we use the embeddings to define a non-local feature density $\boldsymbol{g}_{\text{NL}} : \mathbb{R}^3 \to \mathbb{R}^d$ for the XC energy density $\epsilon_{\text{XC}}$ from Equation 2. Compared to previous approaches, our finite point cloud embeddings are neither dependent on the nuclear charge nor the basis set but purely derived from the density $\rho$. To train on energies alone, we differentiate through the minimization of Equation 1 (Li et al., 2021).

In our experimental evaluation, EG-XC improves upon the learnable semi-local XC-functional on molecular dynamic trajectories, extrapolation to both out-of-distribution conformations and increasingly larger molecules. In particular, we find EG-XC to reduce errors of the semi-local functional by a factor of 2 to 3, on par with accurate ML force fields combined with DFT calculations ($\Delta$-ML). In extrapolation to unseen structures, we find EG-XC to accurately reproduce out-of-distribution potential energy surfaces unlike force fields (incl. $\Delta$-ML) while reducing the MAE by $35\,\%$ to $50\,\%$ compared to the next-best tested method. Finally, we find EG-XC to have an excellent data efficiency, achieving similar accuracies to the best force fields with 5 times less data. To summarize, we demonstrate that GNN-driven functionals push the frontier of non-local XC functionals and provide a promising path toward accurate and scalable DFT calculations.

## 2 BACKGROUND

**Notation.** To denote functionals, i.e., functions mapping from functions to scalars, we write $F[f]$ where $F$ is the functional and $f$ the function. We use superscripts in brackets[(t)] to indicate sequences and regular superscripts of bold vectors, e.g., $\boldsymbol{x}^l$, to indicate the $l$-th irreducible representation of the SO(3) group. We generally use $r \in \mathbb{R}^3$ to denote points in the 3D Euclidean space. A notable exception is the finite set of nuclei positions $\{R_1, \ldots, R_{N_{\text{nuc}}}\}$ for which we will use capital letters. The norm of a vector is given by $\|r\| = \|r\|_2$. For directions, i.e., unit length vectors, we use $\widehat{r} = \frac{r}{\|r\|}$.

**Kohn-Sham density functional theory** is the foundation of our work. Here, we provide a very brief introduction. For more details, we refer the reader to Appendix A or Lehtola et al. (2020). In KS-DFT, the electron density $\rho$ is represented by a set of orthogonal orbitals $\phi_i : \mathbb{R}^3 \to \mathbb{R}$, $\phi_i(x) = C_i^T \boldsymbol{\chi}(x)$, that are defined as linear combinations of a basis set of atomic orbitals $\chi_\mu : \mathbb{R}^3 \to \mathbb{R}$:

$$\rho(r) = \boldsymbol{\phi}(r)^T \boldsymbol{\phi}(r) = \boldsymbol{\chi}(r)^T CC^T \boldsymbol{\chi}(r) = \boldsymbol{\chi}(r)^T P \boldsymbol{\chi}(r) \tag{3}$$

where $P = CC^T \in \mathbb{R}^{N_{\text{bas}} \times N_{\text{bas}}}$ is the so-called density matrix. Given the representation in a finite basis set, one can compute the kinetic energy $T$, external potential $V_{\text{ext}}$, and electron-electron repulsion energies $V_{\text{H}}$ analytically. Unfortunately, such analytical expressions are generally unavailable for $E_{\text{XC}}[\rho]$ as in Equation 2. Thus, one relies on numerical integration (Lehtola et al., 2020).

To minimize Equation 1, one typically uses the self-consistent field (SCF) method that we outline in greater detail in Appendix B. The SCF method is an iterative two-step optimization where one first computes the so-called Fock matrix $F = \frac{\partial E}{\partial P}$ based on the current coefficients $C$. Then, the optimal coefficients $C \in \mathbb{R}^{N_{\text{bas}} \times N_{\text{el}}}$ are obtained by solving the generalized eigenvalue problem

$$FC = SCE \tag{4}$$

---

[1]We provide the source code on https://github.com/eseberhard/eg-ex

with $S_{\mu\nu} = \int_{\mathbb{R}^3} \chi_\mu(r)\chi_\nu(r)dr$ being the overlap matrix of the atomic orbitals and $E$ being a diagonal matrix. The procedure is repeated until convergence of $P$.

**Equivariance** allows defining symmetries of functions. A function $f : \mathcal{X} \to \mathcal{Y}$ is equivariant to a group $\mathcal{G}$ iff $\forall g \in \mathcal{G}, x \in \mathcal{X}, f\left(G_g^{\mathcal{X}}x\right) = G_g^{\mathcal{Y}}f(x)$ where $G_g^{\mathcal{X}}, G_g^{\mathcal{Y}}$ are the representations of $g$ in the domain $\mathcal{X}$ and codomain $\mathcal{Y}$, respectively. Invariance is a special case of equivariance where $G_g^{\mathcal{Y}} = 1$, i.e., $f\left(G_g^{\mathcal{X}}x\right) = f(x)$, for all $g \in \mathcal{G}, x \in \mathcal{X}$.

The real spherical harmonics $Y^l : \mathbb{R}^3 \to \mathbb{R}^{2l+1}, l \in \mathbb{N}_+$, are a well-known example of SO(3)-equivariant functions as they transform under rotation according to the Wigner D matrices $D_R^l \in \mathbb{R}^{(2l+1)\times(2l+1)}, \forall R \in SO(3)$, with $D_R^0 = 1$ being the identity and $D_R^1$ being 3D-rotation matrices:

$$Y^l(D_R^1 x) = D_R^l Y^l(x). \tag{5}$$

Such $2l + 1$-dimensional equivariant functions $h_i : \mathbb{R}^3 \to \mathbb{R}^{2l+1}$, e.g., $h_i = Y^l$, can be recombined using the tensor product

$$(h_1 \otimes h_2)_{m_o}^{l_o} = \sum_{l_1,l_2} \sum_{m_1,m_2} C_{m_o,m_1,m_2}^{l_o,l_1,l_2} h_{m_1}^{l_1} h_{m_2}^{l_2}, \tag{6}$$

where $C \in \mathbb{R}^{(2l_o+1)\times(2l_i+1)\times(2l_j+1)}$ are the Clebsch-Gordan coefficients. As the Wigner D-matrices are orthogonal, it follows that the inner product of two $l$-equivariant functions yields an invariant one

$$h_i^l\left(D_R^1 r\right)^T h_j^l\left(D_R^1 r\right) = h_i^l(r)^T D_R^{lT} D_R^l h_j^l(r) = h_i^l(r)^T h_j^l(r). \tag{7}$$

While, we are interested in $E(3)$-invariant functions, using such equivariant intermediate representations improves accuracy, expands the function class, and improves data efficiency (Schütt et al., 2017; Gasteiger et al., 2021; Batatia et al., 2022).

## 3 RELATED WORK

**Learnable density functional approximations** (DFAs) of the XC functional have a long history. In their seminal paper, Kohn & Sham (1965) proposed an $\epsilon_{\text{XC}}$ fitted to a reference calculation. In this first parameterization $\epsilon_{\text{XC}}$ only depends on $\boldsymbol{g}(r) = \rho(r)$. To improve accuracy, the density features $\boldsymbol{g}$ have been successively refined by including gradients and physical quantities such as the kinetic energy density $\tau(r) = \frac{1}{2}\sum_i^{N_{\text{el}}} |\nabla\phi_i(r)|^2$. The resulting *semi-local* functionals are also known as meta-GGAs with $\boldsymbol{g}(r) = \boldsymbol{g}_{\text{mGGA}}(r) := [\rho(r), |\nabla\rho(r)|, \tau(r)]$ (Perdew, 2001). Many ML functionals use this parameterization with $\epsilon_{\text{XC}}$ frequently being a product of an MLP with a classical functional (Nagai et al., 2022; Kulik et al., 2022; Zhang et al., 2023). An appealing aspect of learning semi-local energy densities rather than $E_{\text{XC}}[\rho]$ directly is the known implementation of physical constraints (Sun et al., 2015; Kaplan et al., 2023), which inspired works by Dick & Fernandez-Serra (2021); Nagai et al. (2022). While augmenting $\boldsymbol{g}_{\text{mGGA}}$ with hand-crafted non-local features improves the accuracy, this comes at the cost of exact constraints (Nagai et al., 2020; Kirkpatrick et al., 2021). Similarly, such physical biasses are absent in kernel methods Margraf & Reuter (2021); Bystrom & Kozinsky (2022). Additionally, these methods require reference densities that are rarely available (Szabo & Ostlund, 2012). Alternative grid-based CNNs converge slowly in the number of grid points compared to specialized spherical integration grids (Zhou et al., 2019; Treutler & Ahlrichs, 1995). Lastly, machine-learned functionals have been used in other DFT contexts as well, e.g., for energy corrections on SCF-converged energies (Chen & Yang, 2024) or to model the kinetic energy functional in orbital-free DFT (Snyder et al., 2012; Mi et al., 2023; Zhang et al., 2024; Remme et al., 2023). Here, we propose to extend semi-local models with expressive fully learnable non-local features obtained from standard integration grids. This allows us to leverage the physical biases of meta-GGAs while allowing efficient non-local interactions with fast integration. Although this work focuses on the XC functional, the methods we present can be transferred to these applications as well.

**Machine learning force fields** have long functioned as a cheap but inaccurate alternative to quantum mechanical calculations by directly approximating the potential energy surface from data without solving the electronic structure problem (Unke et al., 2021). Contemporary force fields generally rely on graph neural networks (GNNs) (Schütt et al., 2017) representing molecules in terms of graphs with atoms as nodes. The advent of SO(3)-equivariant models led to significant improvements in accuracy

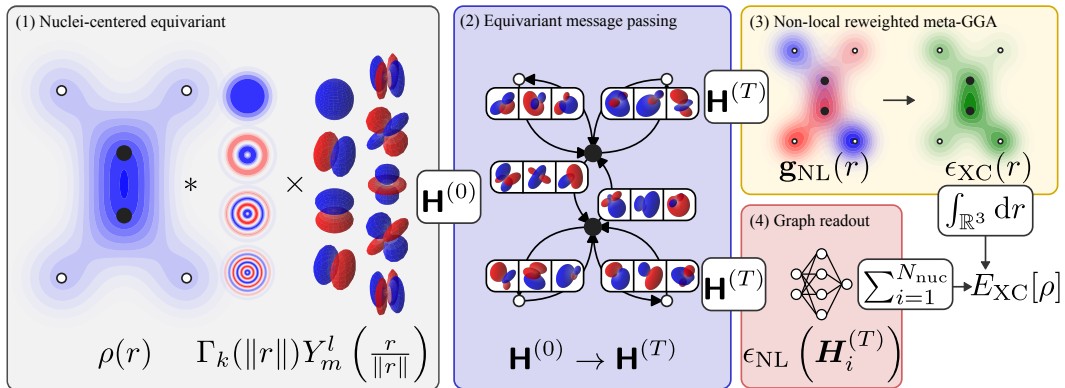

Figure 1: Illustration of Equivariant Graph Exchange Correlation (EG-XC)'s four components: (1) We obtain a finite atomic-range point cloud representation $\mathbf{H}^{(0)}$ by convolving the electron density $\rho$ with radial filters $\Gamma_k : \mathbb{R}_+ \to \mathbb{R}$ and spherical harmonics $Y_m^l : \mathbb{R}^3 \to \mathbb{R}$ at the nuclear position. (2) The embeddings $\mathbf{H}^{(0)}$ are updated using equivariant message passing to obtain molecular-range effects in $\mathbf{H}^{(T)}$. (3) We define a non-local feature density $\boldsymbol{g}_{\mathrm{NL}} : \mathbb{R}^3 \to \mathbb{R}^d$ from which we derive the exchange correlation density $\epsilon_{\mathrm{XC}} : \mathbb{R}^3 \to \mathbb{R}$. (4) We add a graph readout of $\mathbf{H}^{(T)}$ to learn additional corrections. To obtain $E_{\mathrm{XC}}[\rho]$, we integrate $\epsilon_{\mathrm{XC}}$ and add it to the global graph readout.

and data efficiency, closing the gap to full-fidelity quantum mechanical calcualtions (Batzner et al., 2022; Batatia et al., 2022). While they effectively aim to accomplish the same goal as DFT at a fraction of the cost, their out-of-distribution accuracy is a common problem (Stocker et al., 2022). In EG-XC, we transfer the success of equivariant GNNs to DFT, by combining their approximation power with the physical nature of DFT, resulting in an accurate and data-efficient method.

## 4 EQUIVARIANT GRAPH EXCHANGE CORRELATION

With Equivariant Graph Exchange Correlation (EG-XC), we propose an efficient *non-local* approximation to the unknown exchange-correlation functional $E_{\mathrm{XC}}$ that maps electron densities (positive integrable functions in $\mathbb{R}^3$) to scalar energies. To this end, EG-XC consists of four components as illustrated in Figure 1: (1) *Nuclei-centered equivariant embeddings* $\mathbf{H}^{(0)}$ compress the electron density $\rho$ into a finite point cloud representation by equivariantly integrating the density around the nuclei enabling atomic-range interactions. (2) *Equivariant message passing* on this point cloud includes molecular-range information in $\mathbf{H}^{(T)}$. Based on the point cloud embeddings, we define $E_{\mathrm{XC}}$ as the sum of two readouts: (3) A *non-local reweighted meta-GGA* based on our non-local feature density $\boldsymbol{g}_{\mathrm{NL}} : \mathbb{R}^3 \to \mathbb{R}^d$ and (4) a *graph readout* of the invariant nuclei-centered features $\mathbf{H}^{(T)0}$:

$$E_{\mathrm{XC}}[\rho] = \underbrace{\int_{\mathbb{R}^3} \rho(r) \underbrace{\gamma_{\mathrm{NL}}\left(\boldsymbol{g}_{\mathrm{NL}}(r), \boldsymbol{g}_{\mathrm{mGGA}}(r)\right)}_{\text{non-local weights}} \underbrace{\epsilon_{\mathrm{mGGA}}\left(\boldsymbol{g}_{\mathrm{mGGA}}(r)\right)}_{\text{meta-GGA}} dr}_{\text{non-local reweighted meta-GGA}} + \underbrace{\sum_{i=1}^{N_{\mathrm{nuc}}} \epsilon_{\mathrm{NL}}\left(\boldsymbol{H}_i^{(T)0}\right)}_{\text{graph readout}}. \quad (8)$$

**(1) Nuclei-centered equivariant embeddings** $\mathbf{H}^{(0)} \in \left[\mathbb{R}^1 \times \ldots \times \mathbb{R}^{2l_{\max}+1}\right]^{N_{\mathrm{nuc}} \times d} := \mathcal{F}_{l_{\max}}^{N_{\mathrm{nuc}} \times d}$ reduce the computational scaling by mapping the continuous density to a finite per-nucleus representation. Further, these enable using equivariant GNNs to learn molecular-range interactions efficiently. The positions of the nuclei $R_i \in \mathbb{R}^3$ naturally lend themselves as centroids for such embeddings as they represent peaks of the electron density (Kato, 1957). Similar to Remme et al. (2023), we perform this reduction by convoluting the electron density $\rho$ with equivariant filters $S_k^l : \mathbb{R}^3 \to \mathbb{R}^{2l+1}$ at the nuclear positions $R_i$:

$$H_{ik}^{(0)l} = \left(\rho_i * S_k^l\right)(R_i) \quad (9)$$

where $\rho_i : \mathbb{R}^3 \to \mathbb{R}_+$ is the soft-partitioned electron density associated with the $i$-th embedding

$$\rho_i(r) = \rho(r) \frac{\alpha_i(r)}{\sum_{j=1}^{N_{\text{nuc}}} \alpha_j(r)}, \tag{10}$$

$$\alpha_i(r) = \exp\left(-\frac{\|r - R_i\|^2}{\lambda^2}\right) \tag{11}$$

with $\lambda \in \mathbb{R}_+$ being a free parameter. Such partitioning prevents overcounting if nuclei are close, similar to quadrature methods (Becke, 1988). Like equivariant GNNs (Batzner et al., 2022), we define the filters $S_k^l$ as a product of spherical harmonics $Y^l : \mathbb{R}^3 \to \mathbb{R}^{2l+1}$ and radial filters $\Gamma_k : \mathbb{R}_+ \to \mathbb{R}$:

$$S_k^l(r) = \Gamma_k(\|r\|) Y^l(\widehat{r}) \tag{12}$$

where $0 \leq l \leq l_{\text{max}}$ indexes the real spherical harmonics. The spherical harmonics allow us to capture angular changes in the electron density that would otherwise be lost in a purely radial representation. Put together, we compute the nuclei-centered equivariant embeddings

$$
\begin{aligned}
H_{ik}^{(0)l} &= \int_{\mathbb{R}^3} \rho_i(r) \Gamma_k^l(\|r - R_i\|) Y^l\left(\widehat{r - R_i}\right) dr \\
&\approx \sum_{j=1}^{N_{\text{quad}}} w_j \rho(r_j) \alpha_i(r_j) \Gamma_k(\|r_j - R_i\|) Y^l\left(\widehat{r_j - R_i}\right)
\end{aligned}
\tag{13}
$$

with a set of $N_{\text{quad}}$ standard integration points and weights $\{(r_j, w_j)\}_{j=1}^{N_{\text{quad}}} \in \left[\mathbb{R}^3 \times \mathbb{R}\right]^{N_{\text{quad}}}$ (Treutler & Ahlrichs, 1995). Importantly, while these embeddings are centered at the nuclei, they do not embed the nuclear charges, as in ML force fields (Schütt et al., 2017), but the electron density around them. This is important as the derivative with respect to the electron density affects the SCF procedure; see Appendix B. A nuclear charge embedding's derivative would not exist and, thus, not alter the DFT calculation, effectively yielding a force field.

We follow Schütt et al. (2021) and parametrize the radial filters $\Gamma_k$ as a combination of $\sin, \cos$ with a fixed polynomial envelope function $u : \mathbb{R}_+ \to \mathbb{R}$ from Gasteiger et al. (2022) for a cutoff $c \in \mathbb{R}_+$:

$$\Gamma_k(r) = \begin{cases} \sin\left(\frac{r\pi k}{2c}\right) u\left(\frac{r}{c}\right) & \text{if } k \text{ even,} \\ \cos\left(\frac{r\pi(k+1)}{2c}\right) u\left(\frac{r}{c}\right) & \text{if } k \text{ odd.} \end{cases} \tag{14}$$

We found such frequency filters to yield more stable results than Bessel functions as the latter ones are very sensitive close to the centroids (Gasteiger et al., 2022). This sensitivity might be beneficial in force fields but is implicitly given in DFT due to the higher density close to the nuclei.

**(2) Equivariant message passing** allows for the propagation and updating of the equivariant electron density features $\mathbf{H}^{(0)}$ to capture molecular-range dependencies within the electron density like Vander-Waals forces (Frank et al., 2022). The equivariance of these features is important as it allows us to define a non-radial-symmetric feature density $g_{\text{NL}}$ in the next step. We use Batzner et al. (2022)'s NequIP to perform message passing with the SO(3)-equivariant convolution from Thomas et al. (2018) to iteratively update the embeddings $\mathbf{H}^{(t)}$ over $T$ steps:

$$\boldsymbol{H}_i^{(t+1)} = \text{EquiMLP}\left(\text{EquiLin}\left(\boldsymbol{H}_i^{(t)}\right) + \text{EquiConv}\left(\mathbf{H}^{(t)}, \boldsymbol{R}\right)_i\right) + \text{EquiLin}\left(\boldsymbol{H}_i^{(t)}\right), \tag{15}$$

where $\text{EquiLin} : \mathcal{F}_{l_{\text{max}}}^d \to \mathcal{F}_{l_{\text{max}}}^d$ is an equivariant dense layer that mixes features of the same order $l$

$$\text{EquiLin}(\boldsymbol{X})_k = \text{Concat}\left(\left[\sum_{k'=1}^d W_{k'k}^l X_{k'}^l\right]_{l=0}^{l_{\text{max}}}\right). \tag{16}$$

with each $\boldsymbol{W}^l \in \mathbb{R}^{d \times d}$ being a learnable weight matrix. $\text{EquiMLP} : \mathcal{F}_{l_{\text{max}}}^d \to \mathcal{F}_{l_{\text{max}}}^d$ is the equivariant equivalent of a standard MLP, where we use the invariant embeddings $l = 0$ and an activation function $\sigma : \mathbb{R} \to \mathbb{R}$ to gate the equivariant parts $l > 0$:

$$\text{EquiMLP}(\boldsymbol{X}) = \text{EquiLin}(\text{ActEquiLin}(\boldsymbol{X})), \tag{17}$$

$$\text{ActEquiLin}(\boldsymbol{X}) = \text{Concat}\left(\sigma(W^0 \boldsymbol{X}^0), \left[\text{EquiLin}(\boldsymbol{X})^l \circ \sigma\left(W^l \boldsymbol{X}^0\right)\right]_{l=1}^{l_{\text{max}}}\right). \tag{18}$$

Lastly, we define the convolution EquiConv : $\mathcal{F}_{l_{\max}}^{N_{\text{nuc}} \times d} \times \mathbb{R}^{N_{\text{nuc}} \times 3} \to \mathcal{F}_{l_{\max}}^{N_{\text{nuc}} \times d}$ via the tensor product:

$$
\text{EquiConv}(\boldsymbol{X}, \mathbf{R})_t^{l_o} = \sum_{s=1}^{N_{\text{nuc}}} \sum_{l_i, l_f=0}^{l_{\max}} \text{MLP}\left(\Gamma(\|R_s - R_t\|)\right)_{l_o, l_i, l_f}
$$
$$
\cdot \left(\text{EquiMLP}(\boldsymbol{x}_s)^{l_i} \otimes Y^{l_f}\left(\widehat{R_s - R_t}\right)\right)^{l_o}. \tag{19}
$$

To accelerate the computation of the tensor product, we use Passaro & Zitnick (2023)'s efficient equivariant convolutions. For more details, we refer the reader to Batzner et al. (2022).

**(3) Non-local reweighted meta-GGA.** As the majority of the XC energy can be captured by semi-local functionals (Goerigk et al., 2017), starting with a machine-learned meta-GGA $\epsilon_{\text{mGGA}} : \mathbb{R}^{d_{\text{mGGA}}} \to \mathbb{R}$ accounts for most of the XC energy. To correct the meta-GGA for non-local effects, we first define a non-local feature density $\boldsymbol{g}_{\text{NL}} : \mathbb{R}^3 \to \mathbb{R}^d$ based on the non-local embeddings $\left\{\mathbf{H}^{(t)}\right\}_{t=0}^T$:

$$
\boldsymbol{g}_{\text{NL}}(r)_k = \sum_{i=1}^{N_{\text{nuc}}} \alpha_i(r) \sum_{t=0}^{T} \sum_{l=0}^{l_{\max}} \underbrace{Y^l\left(\widehat{r - R_i}\right)^T H_{ik}^{(t)l}}_{\text{angular}} \underbrace{\Gamma(\|r - R_i\|)^T \boldsymbol{w}_k^{(t)l}\left(\boldsymbol{H}_i^{(t)0}\right)}_{\text{radial}} \tag{20}
$$

where $\boldsymbol{w}_k^{(t)l} : \mathbb{R}^d \to \mathbb{R}^d$ are MLPs mapping to radial weights. While the inner product between the spherical harmonics $Y^l$ and our equivariant embeddings $\mathbf{H}^{(t)l}$ expresses angular changes, the inner product of radial basis functions $\Gamma$ and $\boldsymbol{w}_k^{(t)l}$ allows for radial changes. From this feature density $\boldsymbol{g}_{\text{NL}}$ and standard meta-GGA inputs $\boldsymbol{g}_{\text{mGGA}} : \mathbb{R}^3 \to \mathbb{R}^{d_{\text{mGGA}}}$, we derive the non-local correction $\gamma_{\text{NL}} : \mathbb{R}^{d + d_{\text{mGGA}}} \to \mathbb{R}$ to the exchange energy density $\epsilon_{\text{mGGA}}$:

$$
\epsilon_{\text{XC}}(r) = \gamma_{\text{NL}}\left(\boldsymbol{g}_{\text{NL}}(r), \boldsymbol{g}_{\text{mGGA}}(r)\right) \cdot \epsilon_{\text{mGGA}}\left(\boldsymbol{g}_{\text{mGGA}}(r)\right). \tag{21}
$$

In practice, we implement $\gamma_{\text{NL}}$ as an MLP. To obtain the final readout, we integrate the exchange-correlation density over the electron density $\int_{\mathbb{R}^3} \epsilon_{\text{XC}}(r)\rho(r)dr$. Like Equation 13, we evaluate the integral with standard integration grids.

**(4) Graph readout.** In addition to the meta-GGA-based readout, we add global graph readout of the embeddings $\left\{\mathbf{H}^{(t)}\right\}_{t=0}^T$ to capture the remaining non-local effects. We use an MLP $\epsilon_{\text{NL}} : \mathbb{R}^d \to \mathbb{R}$ on top of the invariant embeddings ($l = 0$) to obtain the final exchange-correlation energy:

$$
E_{\text{XC}}[\rho] = \int_{\mathbb{R}^3} \epsilon_{\text{XC}}(r)\rho(r)dr + \sum_{t=0}^{T} \sum_{i=1}^{M} \epsilon_{\text{NL}}\left(\boldsymbol{H}_i^{(t)0}\right). \tag{22}
$$

**Limitations.** While EG-XC demonstrates significant improvements over a semi-local ML functional, it is not free of limitations. First, as we rely on the nuclear positions to represent the electronic density, it is not truly universal, i.e., independent of the external potential $V_{\text{ext}}$ (Kohn & Sham, 1965). Second, the non-local nature of our functional permits no known way to enforce most physical constraints (Kaplan et al., 2023). Though, the importance of these constraints is still debated (Kirkpatrick et al., 2021), see Appendix O. These missing constraints enable the correction of basis set errors through the XC functional. While this may lead to unphysical matches between densities and energies, our experiments suggest that this does not lead to overfitting energies. Third, systems without nuclei, e.g., the homogenous electron gas, cannot be modeled with our approach. In such cases, one may want to replace the real-space point cloud with a frequency representation (Kosmala et al., 2023). Fourth, to handle open-shell systems, e.g., in chemical reactions, one would need to extend the equivariant embeddings to include spin information. Lastly, while more data efficient and better in extrapolation, running KS-DFT is more expensive than a surrogate force field.

## 5 EXPERIMENTS

In the following, we compare EG-XC to various alternative methods of learning potential energy surfaces across several settings. In particular, we focus on the following tasks: interpolating accurate

Table 1: Test set MAE on the CCSD(T) MD17 dataset in m$E_{\mathrm{h}}$. (**best**, second)

| Molecule | Force field | | | $\Delta$-ML | | | KS-DFT | |
|---|---|---|---|---|---|---|---|---|
| | SchNet | PaiNN | NequIP | SchNet | PaiNN | NequIP | Dick | EG-XC |
| Aspirin | 7.01 | 2.82 | 5.52 | 2.02 | 1.20 | 1.04 | 1.94 | **0.69** |
| Benzene | 0.40 | 0.16 | 0.09 | 0.13 | 0.11 | **0.02** | 0.39 | 0.10 |
| Ethanol | 1.41 | 0.89 | 0.95 | 0.93 | 0.42 | 0.25 | 0.85 | **0.21** |
| Malonaldehyde | 2.10 | 1.00 | 2.32 | 0.61 | 0.44 | 0.29 | 0.73 | **0.27** |
| Toluene | 1.80 | 1.10 | 1.87 | 0.44 | 0.31 | **0.13** | 0.38 | 0.20 |

energy surfaces, extrapolation to unseen conformations, and extrapolation to larger molecules. Additionally, we present an ablation study on EG-XC's components. For a runtime complexity analysis, we refer the reader to Appendix M and for runtime measurements to Appendix N.

**Methods.** To accurately position EG-XC, we compare to methods of varying computational costs: force fields, $\Delta$-ML, i.e., combining force fields with KS-DFT calculations, and a learnable XC-functional (Dick & Fernandez-Serra, 2021). While force fields are orders of magnitude cheaper by bypassing quantum mechanical calculations, they lack prior physical knowledge. For the $\Delta$-ML methods, we shift all energies by DFT energies with LDA in the STO-6G basis. This reduces the learning problem to the difference between the KS-DFT and the target energy (Wengert et al., 2021). As force fields, we test increasingly expressive models based on their use of SO(3) irreducible representations: SchNet ($l = 0$) (Schütt et al., 2017), PaiNN ($l = 1$) (Schütt et al., 2021), and NequIP ($l = 2$) (Batzner et al., 2022). Finally, we compare with EG-XC's learnable semi-local XC-functional (Dick & Fernandez-Serra, 2021). As $\Delta$-ML methods and learnable XC-functionals require a DFT computation for each structure, they are at the same computational cost as EG-XC. In Appendix L, we provide additional $\Delta$-ML calculations with learnable XC functionals as base method.

**Setup.** To train the XC functionals, we follow Li et al. (2021) and implement the SCF method differentiably. This allows us to match the converged SCF energies directly to the target energies without needing ground truth electron densities. We provide implementation details in Appendix C. All methods are trained on energy labels only. Force fields are trained with hyperparameters from their respective works with modifications to learning rate, batch size, and initialization to improve performance; we outline the changes in Appendix G. We list EG-XC's hyperparameters in Appendix F.

**Loss.** We follow Dick & Fernandez-Serra (2021) and minimize the mean squared error between the converged SCF energy and the target energy over the last $I_{\mathrm{loss}} \in \mathbb{N}_+$ steps

$$\mathcal{L} = \sum_n \sum_{i=I-I_{\mathrm{loss}}}^{I} \left( E_{\mathrm{target},n} - E_{\mathrm{SCF}}^i(\boldsymbol{R}_n, \boldsymbol{Z}_n) \right)^2 \tag{23}$$

with $E_{\mathrm{target}} \in \mathbb{R}$ beign the target energy and $E_{\mathrm{SCF}}^i : (\mathbb{R}^3 \times \mathbb{N}_+)^M \to \mathbb{R}$ the SCF energy after the $i$-th iteration. The index $n$ runs over the dataset. The parameter gradients are obtained through the total derivative of the loss function with respect to the parameters of the XC functional $\frac{d\mathcal{L}}{d\theta}$ by backpropagating through the SCF iterations. Importantly, for the parameters updates to exist, the XC functional must be at least twice differentiable as we compute the partial derivative $\frac{\partial E_{\mathrm{xc}}[\rho]}{\partial P^i}$ to construct the $i$th Fock matrix (see Appendix B) and the total derivative for the updates $\frac{d\mathcal{L}}{d\theta}$. We obtain a $C^\infty$-smooth XC functionals by using the SiLU activation function (Hendrycks & Gimpel, 2023).

**Reproducing gold-standard accuracies.** The objective behind learning XC-functionals or ML force fields is to facilitate access to accurate potential energy surfaces by distilling highly accurate reference data into a faster method. Hence, we compare these methods on the revised MD17 dataset, which contains precise 'gold-standard' CCSD(T) (CCSD for aspirin) reference energies for conformations of five molecules along the trajectory of a molecular dynamic (MD) simulation (Chmiela et al., 2018). Each molecule has a training set of 1000 structures, which we split into 950 training and 50 validation structures. Each test set contains an additional 500 structures (1000 for ethanol). Following Schütt et al. (2017), a separate model is fitted per molecule. Since CCSD(T) calculations inherently account for non-local effects, these datasets are well suited for investigating the ability to accurately interpolate multi-dimensional energy surfaces from a limited number of reference structures. In Appendix I, we provide additional $\Delta$-ML data with a more accurate DFT functional and basis sets.

Table 2: Structural extrapolation on the 3BPA dataset. All methods were trained on the 300K training set. All numbers are relative MAE in $mE_h$. The last three rows refer to the potential energy surfaces.

| Test set | Force field | | | $\Delta$-ML | | | KS-DFT | |
|---|---|---|---|---|---|---|---|---|
| | SchNet | PaiNN | NequIP | SchNet | PaiNN | NequIP | Dick | EG-XC |
| 300K | 5.15 | 2.91 | 3.81 | 2.38 | 1.14 | 0.81 | 0.96 | **0.42** |
| 600K | 9.06 | 5.81 | 7.55 | 3.96 | 2.13 | 1.56 | 1.36 | **0.73** |
| 1200K | 18.33 | 14.14 | 17.30 | 6.84 | 5.97 | 3.30 | 2.27 | **1.39** |
| $\beta = 120°$ | 3.84 | 1.78 | 2.25 | 2.53 | 1.25 | 1.09 | 0.75 | **0.35** |
| $\beta = 150°$ | 4.64 | 1.89 | 2.64 | 2.03 | 0.84 | 0.88 | 0.61 | **0.23** |
| $\beta = 180°$ | 4.97 | 1.92 | 3.03 | 1.79 | 1.06 | 0.73 | 0.56 | **0.20** |

NequIP   $\Delta$-NequIP   Dick   EG-XC   Target

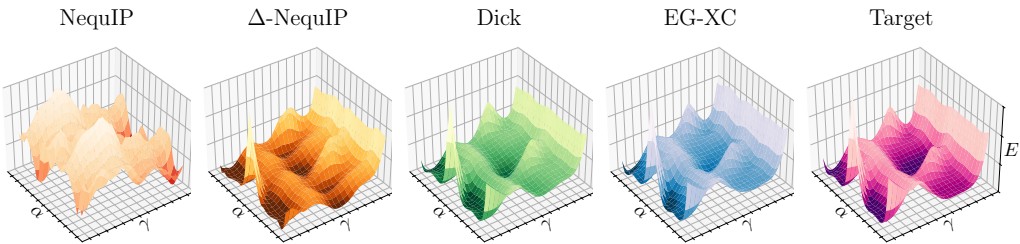

Figure 2: Two-dimensional slice of the potential energy surface of 3BPA with the dihedral $\beta = 120°$. Pure force fields like NequIP struggle to recover the shape of this out-of-distribution energy surface. When paired with DFT calculations, one can see that the energy surface moves closer to the target shape but introduces additional extrema. Learnable XC functionals like Dick & Fernandez-Serra (2021) and EG-XC demonstrate significantly better reproduction of the target energy surface.

Table 1 lists the mean absolute error (MAE) for all methods on the MD17 test sets. We find that force fields generally struggle to reconstruct the energy surface of the MD17 dataset accurately when trained solely on energies. In contrast, $\Delta$-ML methods and learnable XC-functionals generally achieve chemical accuracy at $1 \, \text{kcal} \, \text{mol}^{-1} \approx 1.6 \, mE_h$. The DFT reference calculations systematically reduce the to-be-learned energy surface delta from $6.2 \, mE_h$ to $4.9 \, mE_h$. Compared to the EG-XC's base semi-local Dick & Fernandez-Serra (2021), EG-XC reduces errors by a factor of 2 to 4. Among all methods, EG-XC yields the lowest error on 3 of the 5 molecules. In Appendix H, we test the performance on a reduced training set of just 50 samples. There we find that gap between KS-DFT methods and force fields ones to widen significantly supporting the data efficiency of learnable KS-DFT methods. We hypothesize the good performance of force fields on MD17 being due to the homogeneous nature of the dataset where the gap between the training and test set is small. In such settings, no physical inductive bias may be required to perform well on the test set. Conversely, we expect the performance of force fields to degrade in extrapolation settings where the training and test set differ significantly. We investigate this hypothesis in the following experiments.

**Extrapolating structures.** In MD simulations, one often encounters structures far outside the training set, e.g., due to higher temperatures or environmental changes (Stocker et al., 2022). To investigate the extrapolation to unseen structures, we use the 3BPA dataset (Kovács et al., 2021). 3BPA contains various geometric configurations of the molecule 3-(benzyloxy)pyridin-2-amine. The training set consists of 500 structures sampled from an MD simulation at room temperature (300K). The test sets consist of MD trajectories at 300K, 600K, and 1200K to test in and out-of-distribution (OOD) performance. Additionally, the dataset contains three 2-dimensional potential energy surface slices where two dihedral angles are varied while keeping one fixed. The labels have been computed with the hybrid $\omega$B97X XC functional and the 6-31G(d) basis set and, thus, include non-local interactions.

As constant offsets of the potential energy surface do not affect the system dynamics, we evaluate the relative MAE, i.e., $\mathbb{E}\left[|E_{\text{pred}} - E_{\text{target}} - \text{median}(E_{\text{pred}} - E_{\text{target}})|\right]$. We list the relative MAE for all methods and test sets in Table 2; for the absolute MAE, we refer the reader to Appendix J. Across all test sets, EG-XC results in $35\%$ to $51\%$ lower errors than the next best-tested alternative. On the

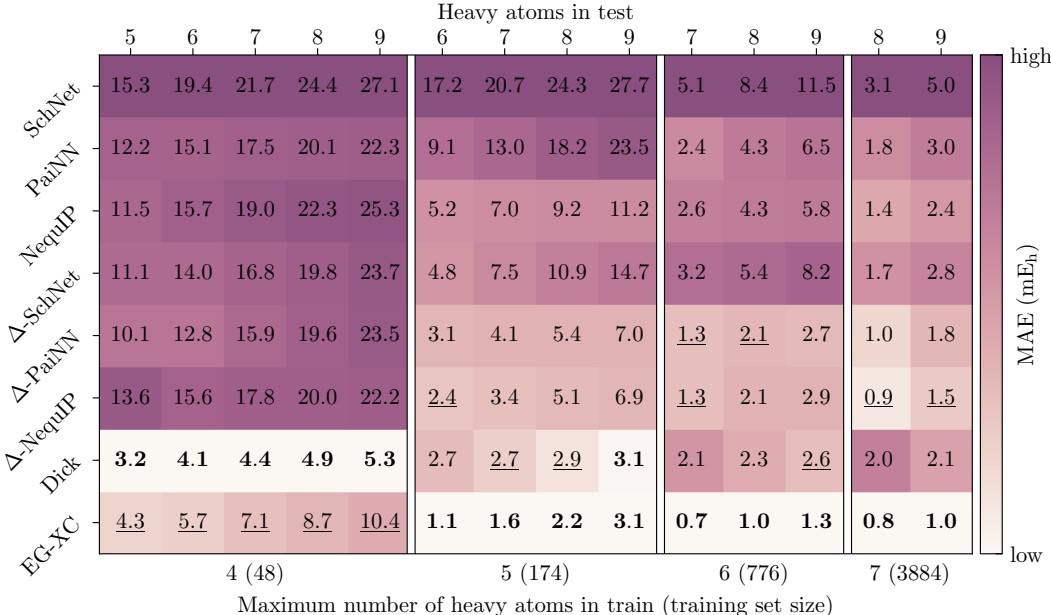

Figure 3: MAE in m$E_{\mathrm{h}}$ on QM9 size extrapolation. Each row represents a different method. The four groups indicate the maximum number of heavy atoms in the training set and its size. Each column represents the test subset with the number of heavy atoms listed above.

far OOD 1200K samples, EG-XC is the only method achieving chemical accuracy $1.6\,\mathrm{m}E_{\mathrm{h}}$ at an relative MAE of $1.40\,\mathrm{m}E_{\mathrm{h}}$. To illustrate the qualitative improvement of EG-XC's energy surfaces, we plot the most OOD potential energy slice at $\beta = 120°$ in Figure 2 for NequIP, $\Delta$-NequIP, Dick & Fernandez-Serra (2021), EG-XC and the target. The remaining methods and corresponding energy surfaces are plotted in Appendix K. It is evident that force fields fail to reproduce the target energy surface with no resemblance to the target surface. While the reference DFT calculations move $\Delta$-NequIP closer to the target surface, the energy shape includes additional extrema not present in the target surface. In contrast, both XC functionals accurately reproduce the energy surface. In line with the results on MD17, we find support to the hypothesis that learnable XC functionals are better suited for extrapolation to unseen structures than force fields.

**Extrapolation to larger molecules.** Gathering reference data for large compounds is costly and expensive with accurate quantum chemical calculations like CCSD(T) scaling $O(N_{\mathrm{el}}^7)$ in the number of electrons $N_{\mathrm{el}}$. Thus, extrapolation from small or medium-sized molecules to larger ones is critical. Here, we simulate this setting by splitting the QM9 dataset (Ramakrishnan et al., 2014) into subsets of increasing size based on the number of heavy atoms, i.e., QM9($S$) are all QM9 molecules with at most $S$ heavy atoms. For each $S \in \{4, 5, 6, 7\}$, we train a separate model and test on the remaining structures, i.e., QM9\QM9($S$). For each training set, we split the structures 90%/10% into training and validation sets. The QM9 dataset comprises 134k stable organic molecules with up to 9 heavy atoms. The energies have been computed with the hybrid B3LYP XC functional with the 6-31G(2df,p) basis set and, thus, contain non-local interactions through the exact Hartree exchange. As the dataset contains only few molecules with fluorine, force fields (including $\Delta$-ML) could not yield accurate energies if none or few are in the training set. Thus, we omitted all molecules that include fluorine.

We visualize the MAE for each combination of the number of heavy atoms in the test set and the maximum number of heavy atoms in the training set in Figure 3. Across all combinations of training and test sets, we find learnable XC functionals yielding the lowest errors with a preference for Dick & Fernandez-Serra (2021) on the smallest QM9(4). We hypothesize that this is due to the physical constraints enforced by the learnable XC functional, which are not present in the other methods, including EG-XC. Starting with QM9(5), EG-XC's errors are consistently the lowest. Notably, EG-XC trained on QM9(6) yields lower MAE on the largest structures than the best alternative on QM9(7) with 5 times more samples and one-atom larger molecules. On QM9(7), EG-XC is at least 33 % more accurate than the competing methods on test molecules with 9 heavy atoms. Compared to Dick & Fernandez-Serra (2021), EG-XC reduces the MAE by at least 2× on QM9(6) and QM9(7).

Table 3: Ablation study on the 3BPA dataset.

|                  | 300K | 600K | 1200K | $\beta = 120°$ | $\beta = 120°$ | $\beta = 120°$ |
|------------------|------|------|-------|----------------|----------------|----------------|
| no mGGA          | 7.00 | 12.89 | 25.85 | 10.99 | 11.16 | 10.82 |
| no graph readout | 0.97 | 1.38  | 2.30  | 0.77  | 0.63  | 0.57  |
| no GNN           | 0.60 | 0.87  | 1.59  | 0.57  | 0.54  | 0.53  |
| EG-XC            | **0.42** | **0.73** | **1.39** | **0.35** | **0.23** | **0.20** |

Overall, the QM9 results support the hypothesis that learnable XC functionals are better suited for extrapolation to larger molecules than force fields.

**Ablations.** To highlight the importance EG-XC's components, we perform an ablation study on the 3BPA dataset. We compare the full EG-XC model to variants without the mGGA, the graph readout, and the equivariant GNN. For the model without GNN, we use an equivariant MLP defined in Equation 17 that we apply node-wise. We report the relative MAE on all test sets in Table 3. One sees that the mGGA is a crucial component for the performance of EG-XC. The model without graph readout approximately doubles EG-XC's error close to the performance of Dick & Fernandez-Serra (2021). Without the equivariant message passing, we observe that EG-XC can still reduce errors significantly, indicating that our equivariant convolution is well-suited to encode the electron density. Otherwise, we would not observe an improvement here as a charge embedding could only correct a constant error, which is accounted for in the relative MAE metric.

# 6 DISCUSSION

We tackled the problem of learning efficient non-local exchange-correlation functionals to enhance the accuracy of density functional theory. To this end, we have presented Equivariant Graph Exchange Correlation (EG-XC), an equivariant graph neural network approach. We have introduced a basis-independent reduction of the electron density into a finite point cloud representation, enabling equivariant graph neural networks to capture molecular-range information. Based on this point cloud embedding, we have parameterized a non-local feature density used for the reweighing of a semi-local exchange-correlation energy density. Unlike other learnable non-local functionals, EG-XC can be trained by differentiating through the SCF solver such that the training requires only energy targets.

In our empirical evaluation, EG-XC improves upon previous machine learning-based methods of modeling potential energy surfaces. On MD17, EG-XC accurately reconstructs 'gold-standard' potential energy surfaces, namely CCSD(T), within the DFT framework. On the 3BPA, EG-XC reduces errors by $35\,\%$ to $50\,\%$ compared to the best-performing baseline. On QM9, EG-XC demonstrates remarkable data efficiency, achieving similar accuracies with 5 times less data or up to $33\,\%$ lower errors at the same amount of data compared to the best baseline. Finally, EG-XC extrapolates well to unseen conformations and larger molecules on 3BPA and QM9, respectively.

Overall, these results strongly underline the data efficiency of learning exchange-correlation functionals compared to machine-learned force fields. We hypothesize that a significant portion of EG-XC's generalization is thanks to including the physically constrained semi-local model that captures most of the exchange-correlation energy and biases EG-XC to approximately fulfill the same constraints. On top of these, our non-local contributions transfer the accuracies of graph neural network-based force fields to functionals while maintaining a similar data efficiency.

**Future work.** The simple implementation of EG-XC through standard deep learning libraries and the integration with the self-consistent field method open the door to a range of future research. Thanks to the basis-set independence, EG-XC can transfer to non-atom-centered basis sets like plane waves, as they are common in periodic systems, or approximate the kinetic energy functional in orbital-free DFT. Further, while we trained EG-XC solely on energies, multimodal training with other DFT-computable observables like electron densities or atomic forces could further improve accuracy and generalization. Another path to improving generalization may be integrating known physical constraints to the non-local part of EG-XC. Finally, accurate functionals like EG-XC may bridge the gap between sparse, accurate quantum mechanical calculations (Cheng et al., 2024; Gao & Günnemann, 2024a) and fast force field (Batzner et al., 2022).

**Acknowledgements.** We are grateful to Leo Schwinn and Jan Schuchardt for their invaluable manuscript feedback and to Arthur Kosmala and Johannes Margraf for insightful discussions on density functional theory. Funded by the Federal Ministry of Education and Research (BMBF) and the Free State of Bavaria under the Excellence Strategy of the Federal Government and the Länder.

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

## A  KOHN-SHAM DENSITY FUNCTIONAL THEORY

In Kohn-Sham density functional theory (KS-DFT) (Kohn & Sham, 1965), one attempts to approximate the ground state energy of a molecular system by finding the electron density that minimizes the energy functional Equation 1. Compared to ab-initio methods like Gao & Günnemann (2022; 2023b;a; 2024b); Wilson et al. (2021; 2023), KS-DFT requires the empirical approximation of the XC functional $E_{XC}$ but scales more favorably with system size. Here, we largely compress Lehtola et al. (2020) to provide a brief introduction to the topic. We will neglect electron spin for simplicity and focus on a discretized DFT scheme based on atom-centered basis sets. Operators, i.e., functions mapping from functions to functions, are denoted by acting upon the ket, e.g., $\nabla|f\rangle$ is the derivative of $f$. Further, we use the braket notation to denote the inner product of two functions $f$ and $g$ as $\langle f|g\rangle = \int g(x)f(x)dx$. We will use Einstein's summation convention, where indices that only appear on one side of an equation are summed over.

The electron density $\rho$ of an $N_{el}$ system is associated with an antisymmetric wave function of a fictitious non-interacting system consisting of $N_{el}$ orthonormal orbitals $\phi_i : \mathbb{R}^3 \to \mathbb{R}$. These orbitals are constructed as linear combinations of $N_{bas}$ a finite basis set $\chi_\mu : \mathbb{R}^3 \to \mathbb{R}$ such that $\phi_i(r) = C_{\mu i}\chi_\mu(r)$. For systems without periodic boundary conditions, these basis sets typically consist of nuclei-centered functions called atomic orbitals. The electron density and pseudo-wave-function are given by

$$\Psi(\mathbf{r}) = \frac{1}{\sqrt{N!}} \det\left[\phi_i(r_j)\right]_{i,j=1}^N,\tag{24}$$

$$\rho(r) = \int \Psi(r, r_2, ..., r_N)^* \Psi(r, r_2, ..., r_N) dr_2...dr_N$$
$$= \phi(r)^T \phi(r) = \chi(r)^T C^T C \chi(r) = \chi(r)^T P \chi(r),\tag{25}$$

where $C \in \mathbb{R}^{N_{bas} \times N_{el}}$ are the orbital coefficients and $P = C^T C \in \mathbb{R}^{N_{bas} \times N_{bas}}$ is the so-called density matrix. This construction ensures that the electron density belongs to the class of densities expressible by antisymmetric wave functions and enables the exact computation of the kinetic energy (Kohn & Sham, 1965)

$$T[\rho] = -\frac{1}{2}\langle \phi_i|\nabla^2|\phi_i\rangle = -\frac{1}{2}C_{\mu i}C_{\nu i}\langle \chi_\mu|\nabla^2|\chi_\nu\rangle = C_{\mu i}C_{\nu i}T_{\mu\nu} = P_{\mu\nu}T_{\mu\nu},\tag{26}$$

making KS-DFT significantly more accurate than contemporary orbital-free DFT.

The external potential in molecular systems is defined by the nuclei positions $\mathbf{R}$ and charges $\mathbf{Z}$

$$V_{ext}[\rho] = -\int \frac{\rho(r)Z_n}{|r-R_n|}dr = -C_{\mu i}C_{\nu i}\int \frac{\chi_\mu(r)Z_n\chi_\nu(r)}{|r-R_n|}dr = C_{\mu i}C_{\nu i}V_{\mu\nu} = P_{\mu\nu}V_{\mu\nu}.\tag{27}$$

The Hartree energy is a mean-field approximation to the electron-electron interaction

$$V_H[\rho] = \frac{1}{2}\iint \frac{\rho(r)\rho(r')}{|r-r'|}drdr' = \frac{1}{2}C_{\mu i}C_{\nu i}C_{\lambda j}C_{\kappa j}\iint \chi_\mu(r)\chi_\lambda(r')\frac{1}{|r-r'|}\chi_\nu(r)\chi_\kappa(r')drdr'$$
$$= C_{\mu i}C_{\nu i}C_{\lambda j}C_{\kappa j}J_{\mu\nu\lambda\kappa} = P_{\mu\nu}P_{\lambda\kappa}J_{\mu\nu\lambda\kappa}.\tag{28}$$

Finally, the exchange-correlation energy is a functional of the electron density, which is unknown and has to be approximated. This is the main problem we are addressing in this work.

To find the ground state, the parameters $C$ are optimized to minimize Equation 1. This is typically achieved iteratively in a self-consistent fashion; see Appendix B. The construction via Gaussian-type basis functions (GTOs) greatly simplifies the integrals appearing in the calculation allowing us to analytically precompute the so-called core Hamiltonian $H_{core \mu\nu} = T_{\mu\nu} + V_{\mu\nu}$ and the electron-electron repulsion tensor $J_{\mu\nu\lambda\kappa}$ once and reuse them throughout the optimization.

## B  SELF-CONSISTENT FIELD METHOD

In this work, we use the Self-Consistent Field (SCF) method (Kohn & Sham, 1965; Lehtola et al., 2020) to find the coefficients $\hat{C}$ that minimize the energy functional $E[\rho(r; C)]$ in Eq. 1

$$\hat{C} = \arg\min_C E = \arg\min_C P_{\mu\nu}H_{core \mu\nu} + P_{\mu\nu}P_{\lambda\kappa}J_{\mu\nu\lambda\kappa} + E_{XC}\tag{29}$$

$$\text{s.t. } \forall_{ij}\langle \phi_i|\phi_j\rangle = \delta_{ij}.\tag{30}$$

We can express the orthogonality constraint above via the overlap matrix $S \in \mathbb{R}^{N_{\text{bas}} \times N_{\text{bas}}}$

$$\forall_{ij} \langle \phi_i | \phi_j \rangle = \delta_{ij} \iff C^T S C = I, \tag{31}$$

$$S_{\mu\nu} := \langle \chi_\mu | \chi_\nu \rangle. \tag{32}$$

This constraint is enforced by introducing the Lagrange multipliers $E_{ij}$, and we define the loss

$$\mathcal{L} = E + (C^T S C - I)_{ij} E_{ij}. \tag{33}$$

Since the matrix $E$ is symmetric and the rotation of the basis does not change the energy, we can choose $E$ to be diagonal. Using the Fock matrix $F := \frac{\partial E}{\partial P}$, we can write the gradient of the loss w.r.t. $C$ as

$$\frac{\partial \mathcal{L}}{\partial C} = 2FC - 2SCE \stackrel{!}{=} 0 \tag{34}$$

$$\implies FC = SCE \tag{35}$$

where the Fock matrix is given by

$$F_{\mu\nu} = H_{\text{core}\,\mu\nu} + 2P_{\lambda\kappa} J_{\mu\nu\lambda\kappa} + \frac{\partial E_{\text{XC}}[\rho]}{\partial P_{\mu\nu}}. \tag{36}$$

To evaluate the $E_{\text{XC}}$ and its derivative we discretize $\mathbb{R}^3$ with spherical integration grids around the nuclei (Becke, 1988). Specifically, for a set of quadrature points $\{r_i\}_{i=1}^{N_{\text{quad}}} \in \mathbb{R}^{N_{\text{quad}} \times 3}$ and weights $\{w_i\}_{i=1}^{N_{\text{quad}}} \in \mathbb{R}_+^{N_{\text{quad}}}$, we first compute the density and mGGA features $\boldsymbol{g}_{\text{mGGA}}(r_i) = [\rho(r_i), \|\nabla \rho(r_i)\|, \tau(r_i)]$ on these points:

$$\rho(r_i) = \boldsymbol{\chi}(r_i)^T P \boldsymbol{\chi}(r_i), \tag{37}$$

$$\|\nabla \rho(r_i)\| = 2\|\boldsymbol{\chi}(r_i)^T P \nabla \boldsymbol{\chi}(r_i)\|_2, \tag{38}$$

$$\tau(r_i) = \frac{1}{2} \nabla \boldsymbol{\chi}(r_i)^T P \nabla \boldsymbol{\chi}(r_i). \tag{39}$$

From this discretized input, we compute the scalar output $E_{\text{XC}}$, e.g., $E_{\text{XC}}[\rho] = \sum_{i=1}^{N_{\text{quad}}} w_i \rho(r_i) \epsilon_{\text{mGGA}}(\boldsymbol{g}_{\text{mGGA}}(r_i))$ for an mGGA, and rely on automatic differentiation via backpropagation to compute the derivative w.r.t. $P$. The generalized eigenvalue problem in Equation 35 is solved to find the coefficients $C$. We solve this by diagonalizing $S = V\Lambda V^T$, substituting $C = X\tilde{C}$ with $X = V\Lambda^{-1/2}V^T$, and multiplying both sides from the left with $X^T$ to obtain the ordinary eigenvalue problem

$$\underbrace{X^T F X}_{\tilde{F}} \tilde{C} = \underbrace{X^T S X}_{I} \tilde{C}E$$
$$\implies \tilde{F} = \tilde{C}E. \tag{40}$$

After solving this for $\tilde{C}$, we recover the original coefficients $C = X^T \tilde{C}$ via the eigenvectors associated with the $N_{\text{el}}$ lowest eigenvalues $E$. Since $F$ depends on $P = CC^T$, this problem is linearized by alternating optimization steps of $F(C)$, $C$, s.t. we need to iterate until convergence commonly referred to as self-consistence in this context.

## C  SELF-CONSISTENT FIELD IMPLEMENTATION

We implement the SCF method from Appendix B fully differentiably in JAX (Bradbury et al., 2018) by following Lehtola et al. (2020). In particular, we precompute $\boldsymbol{\chi}(r)$ on the integration grid points (Treutler & Ahlrichs, 1995), and the core-Hamiltonian $H_{\mu\nu}$. Since the electron-repulsion integrals $J_{\mu\nu\lambda\kappa}$ scale as $O(N_{\text{bas}}^4)$ in compute and memory, we use the density-fitting approximation (Vahtras et al., 1993) to reduce the memory and compute scaling to $O(N_{\text{bas}}^2 N_{\text{aux}})$ where $N_{\text{aux}} \ll N_{\text{bas}}^2$ is the size of the auxiliary basis set. We explain the procedure in more detail in Appendix D. Additionally, we improve the convergence by implementing the direct inversion of the iterative subspace (DIIS) method, which we briefly summarize in Appendix E (Pulay, 1982). For precomputing the integrals and obtaining grid points, we use the PySCF (Sun et al., 2018). For the evaluation of the atomic orbitals on the integration grid points, we use our own JAX implementation.

Like Dick & Fernandez-Serra (2021), we perform several SCF iterations with a pre-defined XC-functional to obtain a good initial guess for the density. During training, we randomly interpolate between the precycled initial guess and a standard initial guess via

$$P_0 = \frac{1+\alpha}{2} P_{\text{precycle}} + \left(1 - \frac{1+\alpha}{2}\right) P_{\text{standard}} \tag{41}$$

where $\alpha \sim \mathcal{U}(0,1)$. At inference time, we fix $\alpha = 1$. This interpolation leads to varying initial densities and functions as a regularizer (Dick & Fernandez-Serra, 2021).

## D    DENSITY FITTING

Here, we largely follow Krisiloff et al. (2015) and give a brief introduction to the density-fitting approximation. For more details, we refer the reader to the original work. We want to approximate the two-electron integral tensor

$$J_{\mu\nu\lambda\kappa} = \int \chi_\mu(r)\chi_\nu(r')\frac{1}{|r-r'|}\chi_\lambda(r)\chi_\kappa(r')drdr'. \tag{42}$$

To reduce the memory and compute scaling from $O(N_{\text{bas}}^4)$ to $O(N_{\text{bas}}^2 N_{\text{aux}})$, we introduce an auxiliary basis set $\{\chi_\mu^{\text{aux}}\}_{\mu=1}^{N_{\text{aux}}}$ and approximate the two-electron integrals as

$$J_{\mu\nu\lambda\kappa} \approx \hat{J}_{(\mu\nu)I}\tilde{J}_{IJ}^{-1}\hat{J}_{(\lambda\kappa)J} \tag{43}$$

where

$$\hat{J}_{(\mu\nu)I} = \int \chi_\mu(r)\chi_\nu(r)\frac{1}{|r-r'|}\chi_I^{\text{aux}}(r)drdr', \tag{44}$$

$$\tilde{J}_{IJ} = \int \chi_I^{\text{aux}}(r)\frac{1}{|r-r'|}\chi_J^{\text{aux}}(r')drdr', \tag{45}$$

and the indices $I, J$ run over the auxiliary basis set $\{\chi_I^{\text{aux}}\}_{I=1}^{N_{\text{aux}}}$. Note that choosing $\{\chi_I^{\text{aux}}\}_{I=1}^{N_{\text{aux}}} = \{\chi_\mu \cdot \chi_\nu\}_{\mu,\nu=1}^{N_{\text{bas}}}$ yields the exact two-electron integrals. One can further simplify by computing

$$B_{(\mu\nu)I} = \hat{J}_{(\mu\nu)J}\tilde{J}_{IJ}^{-\frac{1}{2}} \tag{46}$$

where $\tilde{J}^{-\frac{1}{2}} = \tilde{J}^{-1}\tilde{J}^{\frac{1}{2}}$ with $\tilde{J}^{\frac{1}{2}}$ being the Choleksy decomposition of $\tilde{J}$. This reduces the contraction of the two-electron integrals with the density matrix to

$$V_{\text{H}}[\rho] = P_{\mu\nu}J_{\mu\nu\lambda\kappa}P_{\lambda\kappa} \approx P_{\mu\nu}B_{(\mu\nu)I}B_{(\lambda\kappa)I}P_{\lambda\kappa}. \tag{47}$$

## E    DIRECT INVERSION OF ITERATIVE SUBSPACE

The direct inversion of the iterative subspace (DIIS) method is a common technique to accelerate the convergence of the SCF method. Here, we briefly summarize the method; for a derivation and more details, we refer the reader to Pulay (1982). The DIIS method aims at finding a linear combination of previous Fock matrices that minimizes the norm of the error matrix. This accelerates the convergence of the SCF method by extrapolating a new Fock matrix from previous ones. Let $F_k$ be the Fock matrix in the $k$-th iteration and $P_k$ the density matrix. Then the error matrix is defined as

$$e_k = \left(S^{-\frac{1}{2}}\right)^T (F_k P_k S - S P_k F_k)S^{-\frac{1}{2}} \tag{48}$$

where $S$ is the overlap matrix. We aim to find coefficients $c_i$ that minimize the norm of a linear combination of error matrices

$$\min_{\{c_i\}}\left\|\sum_{i=1}^m c_i e_i\right\|^2 \tag{49}$$

subject to the constraint $\sum_{i=1}^m c_i = 1$. This minimization problem can be solved using Lagrange multipliers, leading to the linear system

$$\begin{pmatrix} \boldsymbol{B} & \boldsymbol{1} \\ \boldsymbol{1}^T & 0 \end{pmatrix}\begin{pmatrix} \boldsymbol{c} \\ \lambda \end{pmatrix} = \begin{pmatrix} \boldsymbol{0} \\ 1 \end{pmatrix} \tag{50}$$

where $B_{ij} = \text{Tr}(e_i^T e_j)$, $\boldsymbol{1}$ is a vector of ones, and $\lambda$ is a Lagrange multiplier. Solving this system gives us the optimal coefficients $c_i$ which we use to obtain the extrapolated Fock matrix that is solved in the next SCF iteration.

Table 4: Hyperparameters for EG-XC.

| Hyperparameter | Value |
|---|---|
| $d$      number of features per irrep | 32 |
| $l_{\max}$    number of irreps | 2 |
| $T$       number of layers | 3 |
| Radial filters | 32 |
| $\epsilon_{\mathrm{mGGA}}$ Base semilocal functional | Dick & Fernandez-Serra (2021) |
| Batch size | 1 |
| $I_{\mathrm{loss}}$    Number of steps to compute loss | 3 |
| Parameter EMA | 0.995 |
| Optimizer | Adam |
| $\beta_1$ | 0.9 |
| $\beta_2$ | 0.999 |
| Basis set | 6-31G(d) |
| Density fitting basis set | weigend |
| $I$     SCF iterations | 15 |
| Precycle XC functional | LDA |
| Precycle iterations | 15 |
| Learning rate | |
|    MD17 | $\frac{0.01}{1+\frac{1}{1000}}$ |
|    3BPA | $\frac{0.01}{1+\frac{1}{1000}}$ |
|    QM9 | $\frac{0.001}{1+\frac{1}{1000}}$ |

## F   HYPERPARAMETERS

We list the hyperparameters for EG-XC in Table 4.

## G   BASELINE CHANGES

For reference methods, we used the model hyperparameters from their respective works. For NequIP, we used the default $l = 2$. We extended patience schedules to ensure full convergence for all baselines. On the QM9($S$) datasets, we first fitted a linear model on the training set with the number of atoms and shifted all labels by this prediction. Further, we initialized all models with zero in the last layer. These steps improved generalization between 10 and 100 times on the smaller datasets.

## H   MD17 WITH 50 SAMPLES

We imitate the setting of Batatia et al. (2022) and train all models on only 50 samples for each of the MD17 trajectories. We reduce the validation set to 10 samples as well. As we found KS-DFT methods to be more learning rate sensitive here, we ablated the initial learning rate with 0.01 and 0.001 and report the lower ones for Dick & Fernandez-Serra (2021) and EG-XC. The resulting test set MAEs are listed in Table 5. KS-DFT methods and $\Delta$-ML are more successful learning from such few samples. Especially on the challenging aspirin structures, EG-XC is the only method yielding accuracies within chemical accuracy.

## I   DELTA ML WITH SCAN (6-31G) DFT

In Table 6, we present additional MD17 data with the SCAN functional (Sun et al., 2015) and the 6-31G(d) basis set. The results show that improving the reference DFT functional significantly improves $\Delta$-ML approaches. It should be noted that the pure SCAN is already more accurate than the Dick & Fernandez-Serra (2021) specifically fitted functional on Aspirin. We argue that this setting is

Table 5: MD17 with only 50 samples.

| | Force field | | | Δ-ML | | | KS-DFT | |
|---|---|---|---|---|---|---|---|---|
| | SchNet | PaiNN | NequIP | SchNet | PaiNN | NequIP | Dick | EG-XC |
| Aspirin | 8.94 | 9.46 | 8.75 | 9.55 | 3.27 | 3.19 | 1.98 | **1.23** |
| Benzene | 2.11 | 2.28 | 2.84 | 0.25 | 0.26 | **0.21** | 0.53 | 0.59 |
| Ethanol | 4.15 | 3.30 | 4.44 | 2.99 | 1.60 | 1.32 | 0.99 | **0.35** |
| Malonaldehyde | 5.47 | 3.99 | 5.05 | 2.81 | 1.77 | 1.36 | 0.78 | **0.54** |
| Toluene | 6.65 | 4.40 | 5.15 | 1.15 | 0.79 | 0.80 | **0.70** | 0.99 |

Table 6: MAE for Δ-ML with SCAN and the 6-31G(d) basis set on MD17. For SCAN, we report the relative MAE instead to account for mean shifts.

| Molecule | SchNet | PaiNN | NequIP | SCAN |
|---|---|---|---|---|
| Aspirin | 0.57 | 0.46 | 0.34 | 1.84 |
| Benzene | 0.08 | 0.07 | 0.02 | 1.27 |
| Ethanol | 0.26 | 0.17 | 0.08 | 1.04 |
| Malonaldehyde | 0.32 | 0.25 | 0.13 | 1.26 |
| Toluene | 0.21 | 0.17 | 0.08 | 1.92 |

an ideal case of Δ-ML as SCAN is well-suited for such close-to-equilibrium structures. However, it should be mentioned that such improvements are orthogonal as one may also apply Δ-learning on top of EG-XC.

## J  3BPA MEAN ABSOLUTE ERROR

In Table 7, we present the absolute MAE on the 3BPA dataset for all methods. It is apparent that while EG-XC performs well, Δ-NequIP achieves the lowest MAE. Thus, demonstrating that EG-XC's error is largly a constant offset that would not affect actual MD simulations.

## K  3BPA POTENTIAL ENERGY SURFACES

We plot the potential energy surfaces for 120°, 150°, and 180° in Figure 4, Figure 5, and Figure 6, respectively. It is apparent that force fields struggle on all energy surfaces while XC-functionals yield a similar surface structure to the target.

Table 7: Absolute MAE in $mE_h$ for structural extrapolation on the 3BPA dataset. All methods are trained on the 300K training set.

| | Force field | | | Δ-ML | | | KS-DFT | |
|---|---|---|---|---|---|---|---|---|
| Test set | SchNet | PaiNN | NequIP | SchNet | PaiNN | NequIP | Dick | EG-XC |
| 300K | 5.15 | 2.91 | 3.85 | 2.38 | 1.14 | 0.81 | 0.96 | **0.42** |
| 600K | 28.14 | 13.85 | 24.56 | 4.09 | 2.13 | 1.56 | 2.19 | **1.43** |
| 1200K | 93.50 | 49.14 | 81.00 | 7.05 | 6.01 | 3.30 | 6.32 | 4.39 |
| $\beta = 120°$ | 19.98 | 4.16 | 13.77 | 2.54 | 1.26 | 1.09 | 1.58 | **1.04** |
| $\beta = 150°$ | 20.13 | 6.64 | 15.53 | 2.59 | 0.89 | **0.88** | 1.60 | 1.02 |
| $\beta = 180°$ | 20.25 | 9.01 | 17.35 | 1.93 | 1.15 | **0.77** | 1.70 | 1.04 |

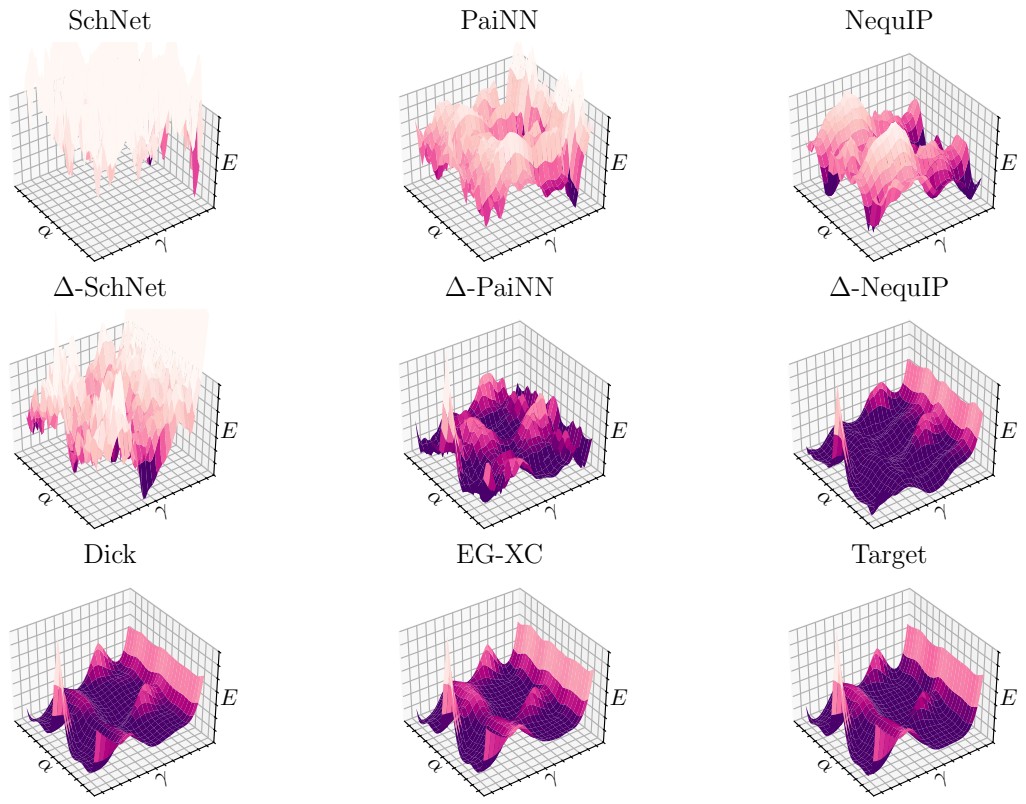

Figure 4: Energy surfaces of the 3BPA dataset at $\beta = 120°$.

Table 8: Relative MAE in $\mathrm{m}E_{\mathrm{h}}$ on the 3BPA dataset with $\Delta$-ML on top of learnable XC functionals.

| Test set | Dick SchNet | PaiNN | NequIP | EG-XC SchNet | PaiNN | NequIP |
|---|---|---|---|---|---|---|
| 300K | 0.62 | 0.30 | 0.34 | 0.42 | **0.29** | 0.30 |
| 600K | 1.01 | 0.56 | 0.62 | 0.71 | **0.52** | 0.56 |
| 1200K | 1.85 | 1.24 | 1.38 | 1.38 | **1.15** | 1.26 |
| $\beta = 120°$ | 0.39 | 0.28 | 0.32 | 0.31 | **0.19** | 0.22 |
| $\beta = 150°$ | 0.41 | 0.35 | 0.27 | 0.24 | **0.19** | 0.22 |
| $\beta = 180°$ | 0.45 | 0.37 | 0.25 | 0.20 | **0.15** | 0.21 |

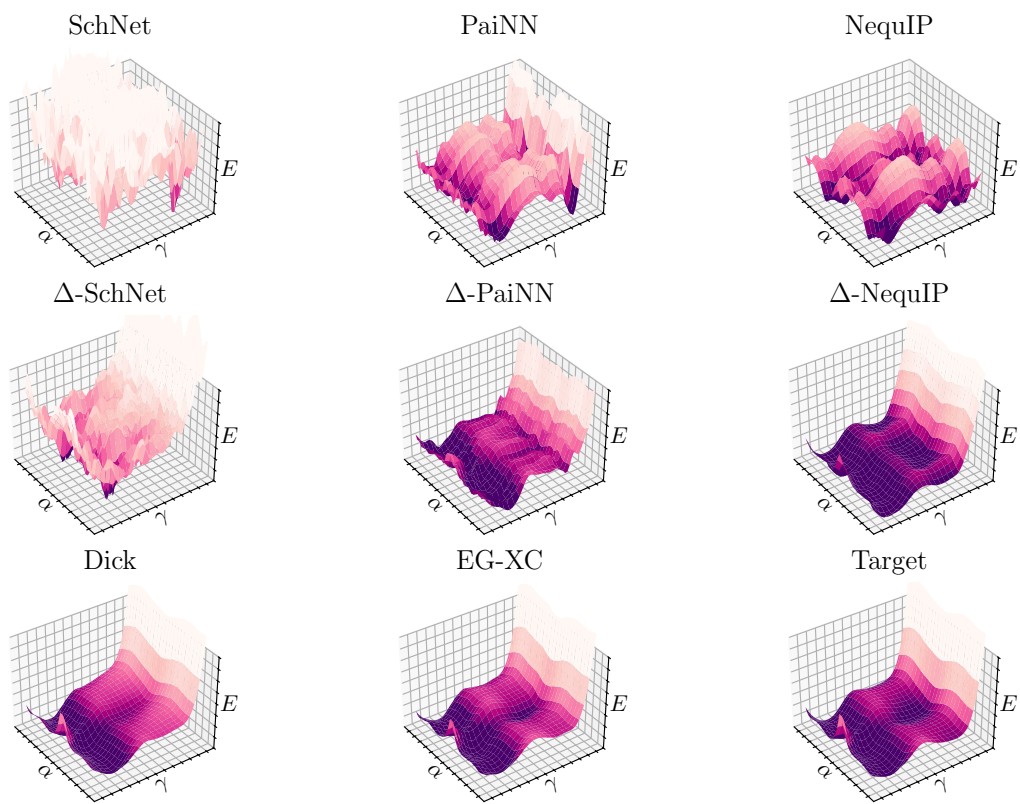

Figure 5: Energy surfaces of the 3BPA dataset at $\beta = 150°$.

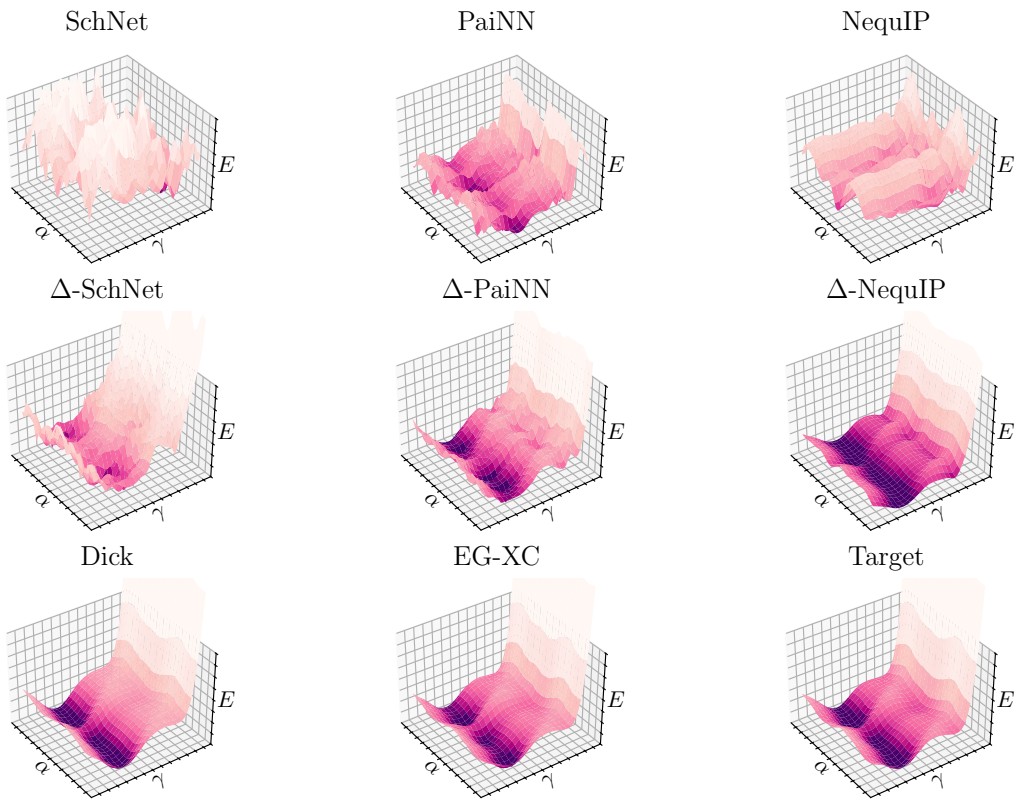

Figure 6: Energy surfaces of the 3BPA dataset at $\beta = 180°$.

## L Δ-LEARNING ON LEARNABLE XC FUNCTIONALS

As one could see in the experiments of the main body, $\Delta$-ML is a powerful technique to boost the accuracy of force fields by integrating quantum mechanical calculations. One may improve this procedure further by relying on learnable XC functionals as baseline method instead of fixed functionals. Here, we investigate this approach by training $\Delta$-ML on top of Dick & Fernandez-Serra (2021) and EG-XC on the 3BPA dataset.

The results are presented in Table 8. One can see that $\Delta$-ML on top of such accuracy XC functionals improves accuracies significantly beyond the $\Delta$ learning with LDA functional in the main body. Across all force field models, we find EG-XC to yield lower errors on all test sets than the base mGGA functional from Dick & Fernandez-Serra (2021). This supports the hypothesis that the improvements learned by EG-XC are not identical with those of $\Delta$-ML.

## M RUNTIME COMPLEXITY

In the following, we ignore the cost of precomputing the necessary integral tensors, the core Hamiltonian $H_{core}$, the density fitting tensor $B$ (see. Appendix D), the evaluation of the atomic orbitals on the quadrature grid, and the initial guess $P_0$ for the density as these do not contribute significantly to the overall runtime. A single EG-XC forward pass consists of the following steps:

1. $O(N_{quad}N_{bas}^2)$: Compute the electron density features $g_{mGGA}$ from the density matrix $P$.
2. $O(N_{quad}dl_{max})$: Compute the nuclear-centered point cloud embeddings $\mathbf{H}$.
3. $O(N_{nuc}^2 dl_{max}^3 T) + O(N_{nuc}d^2 l_{max}T)$: Equivariant message passing on the point cloud embeddings.
4. $O(N_{quad}dl_{max})$: Compute the non-local feature density $g_{NL}$ on the nuclear grid points.
5. $O(N_{quad}d^2 L)$: Compute the XC energy density $\epsilon_{XC}$ from the non-local feature density.
6. $O(N_{nuc}d^2)$: Graph readout to obtain the XC energy.

Note that due to the exponential decay in the partitioned density $\rho_i$, we truncate the summations in Equation 13 and Equation 20 to avoid a $O(N_{quad}N_{nuc})$ scaling. Additionally, a single SCF requires the following steps:

1. $O(N_{bas}^2 N_{aux})$: Compute the electron repulsion integrals.
2. $O(N_{bas}^2)$: Compute the Fock matrix.
3. $O(N_{bas}^3) + O(I^3)$: DIIS extrapolation.
4. $O(N_{bas}^3)$: Solve the generalized eigenvalue problem.
5. $O(N_{bas}^3)$: Compute the density matrix.

For hybrid functionals like B3LYP or $\omega$B97X, one has to add the cost of the exact exchange term with density fitting $O(N_{bas}^3 N_{aux})$.

For a complete SCF calculation, we iterate these steps until convergence which typically requires $I \approx 15$ iterations. In practice, computing the electron density features dominates the runtime for semi-local functionals. For hybrid functionals, the cost of the exact exchange term dominates the runtime.

## N RUNTIME COMPARISON

Here, we compare the runtime of a DFT calculation with different XC functionals, namely

- LDA: a local XC functional,
- SCAN: a mGGA XC functional,
- B3LYP: a hybrid XC functional,

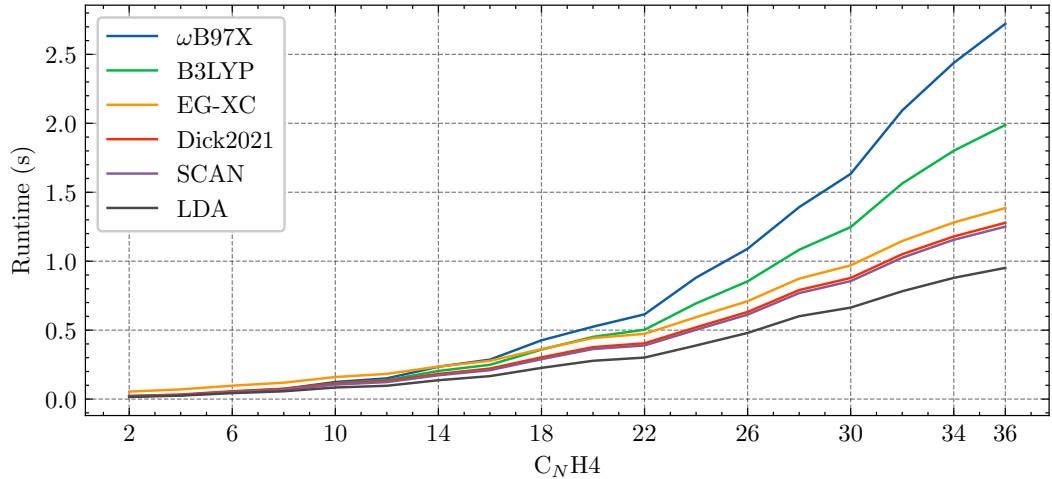

Figure 7: Runtime scaling of KS-DFT methods with different XC functionals.

Table 9: Comparison between Dick & Fernandez-Serra (2021) and Nagai et al. (2020) on 3BPA.

| Test set | Dick | Nagai |
|----------|------|-------|
| 300K | 0.96 | **0.91** |
| 600K | 1.36 | **1.24** |
| 1200K | 2.27 | **2.09** |
| $\beta = 120$ | **0.75** | 0.85 |
| $\beta = 150$ | **0.61** | 0.68 |
| $\beta = 180$ | **0.56** | 0.66 |

- $\omega$B97X: a range-separated hybrid XC functional,
- Dick & Fernandez-Serra (2021): a machine-learned mGGA XC functional,
- EG-XC.

As test system, we use increasing cumulene chains $C_N H_4$ with $N \in \{2, 4, 6, \ldots, 38\}$. We use the 6-31G(d) basis set. We run 15 SCF cycles for each DFT calculation and repeat the calculations 10 times from which we report the minimum to account for other processes running on the same machine. All calculations were performed on a single NVIDIA A100 GPU with our JAX implementation. For the hybrid functionals, we use density fitting to fit the exact exchange term into GPU memory.

The runtime scaling for each method can be found in Figure 7. While DFT calculations are approximately 2 times slower than Dick & Fernandez-Serra (2021) on small structures, at the largest tested system size of $C_{38}H_4$, 232 electrons, EG-XC is only 7 % slower. Compared to hybrid functionals, we find B3LYP and $\omega$B97X being 43 % and 96 % slower than EG-XC, respectively. This highlights the efficiency of EG-XC's non-local interactions in large systems.

## O    PHYSICAL CONSTRAINTS IN mGGA FUNCTIONALS

To investigate the importance of physical constraints on the mGGA, we train Dick & Fernandez-Serra (2021) and Nagai et al. (2020) on the 3BPA dataset. While Dick & Fernandez-Serra (2021) implements many physical constraints, none of these are present in Nagai et al. (2020).

The relative MAE across all test sets is shown in Table 9. While Nagai et al. (2020) yields lower errors on the MD trajectories at higher temperature, Dick & Fernandez-Serra (2021) offers lower errors on the potential energy surfaces. These inconclusive results suggest that physical constraints do not necessarily improve the accuracy of XC functionals, even in extrapolation tasks.

