# OpenReview forum: "Learning Equivariant Non-Local Electron Density Functionals"
_ICLR.cc/2025/Conference — ICLR 2025 Spotlight_

### Official Review · Reviewer_Xv1K · 2024-10-22

**Soundness:** 4
**Presentation:** 3
**Contribution:** 3
**Rating:** 8
**Confidence:** 4

**Summary:**

The introduction of the Global Graph Exchange Correlation functional is innovative, utilizing equivariant graph neural networks to model non-local contributions to the exchange-correlation functional. This approach is relatively uncommon in traditional density functional theory, showcasing the potential of integrating machine learning with quantum chemistry.

**Strengths:**

By combining semi-local functionals with a non-local feature density, GG-XC significantly improves the capture of long-range interactions. Its impressive performance in reconstructing 'gold-standard' CCSD(T) energies highlights its effectiveness in enhancing computational accuracy.

**Weaknesses:**

The article offers a limited background overview and would benefit from a more thorough discussion of certain concepts. For instance, the section on 'non-local functionals' lacks detail and should clarify their significance in density functional theory.

Furthermore, the claim that 'the equivariant GNN efficiently captures long-range information' is somewhat misleading, as many equivariant graph neural networks (EGNNs) are designed with the assumption that inter-atomic interactions are localized. The underlying message-passing scheme typically aggregates information only from atoms within a cutoff radius, which raises questions about their ability to effectively model long-range interactions.

Finally, a comparison of the computational complexity of different methods is necessary.

**Questions:**

1. The claim that 'the equivariant GNN efficiently captures long-range information' is somewhat misleading, as many equivariant graph neural networks (EGNNs) are designed with the assumption that inter-atomic interactions are localized. The underlying message-passing scheme typically aggregates information only from atoms within a cutoff radius, which raises questions about their ability to effectively model long-range interactions.

2. Finally, a comparison of the computational complexity of different methods is necessary.

---

> ### Author Response · Authors · 2024-11-21
>
> We highly appreciate that the reviewer is impressed by the accuracy of our functional. We would like to address the reviewer's concerns with our revised manuscript and the following answer.
>
> **W1: background**
>
> We strive to effectively communicate with both the natural science and the machine learning community. Based on the current feedback, we made the following improvements:
> 1. The introduction includes a proper definition of non-local functionals in the introduction:
>     > While [semi-local functionals] integrate well into existing quantum chemistry code, non-local interactions like Van der Waals forces exceed the functional class (Kaplan et al., 2023). In contrast, *non-local* functionals can capture such interactions by depending on multiple points in space simultaneously, e.g., $E_\text{XC}[\rho]=\iint_{R^3} \rho(r)\rho(r')\epsilon(g(r),g(r'))drdr'$.
> 2. We extended Appendix B with a definition of the Fock matrix $F$ and how it is computed.
> 3. We added a loss definition and gradient computation to the experimental section.
> 4. We use *atomic-range* and *molecular-range* instead of long-range, see next reply.
>
> We highly appreciate any additional feedback on unclear sections.
>
> **W2 + Q1: long-range terminology**
>
> We understand the difficulty in terminology as the term long range is differently used in the force field literature. Given the definition of non-locality above, a non-local functional still does not necessarily need to depend on distant points. To increase clarity and avoid confusion with the force field terminology, we introduce the following two ranges:
> * **atomic-ranged**: electronic interactions around a nucleus.
> * **molecular-ranged**: electronic interactions at typical molecular length scales.
>
> While the convolution with our equivariant filters yields non-local features with atomic-range interactions, molecular-range interactions are added through the equivariant message passing. We adjusted the manuscript accordingly. In a new ablation study on 3BPA, we demonstrate the importance of the molecular-ranged interactions:
> | Metric             | Dick - semi local | EG-XC - atomic ranged (no GNN) | EG-XC - molecular ranged |
> |--------------------|------|----------------|-------|
> | 300K               | 0.96 | 0.60           | **0.42**  |
> | 600K               | 1.36 | 0.87           | **0.73**  |
> | 1200K              | 2.27 | 1.59           | **1.39**  |
> | $\beta=120°$       | 0.75 | 0.57           | **0.35**  |
> | $\beta=150°$       | 0.61 | 0.54           | **0.23**  |
> | $\beta=180°$       | 0.56 | 0.53           | **0.20**  |
>
> This demonstrates that the atomic-range interactions through our point cloud embedding and the molecular-range interactions through our equivariant message passing reduce errors.
>
>
> To further improve our communication, we renamed our method from Global Graph Exchange Correlation (GG-XC) to Equivariant Graph Exchange Correlation (EG-XC), as the interactions may not be global for large structures.
>
>
> **W3 + Q2: Computational complexity**
>
> We agree with the reviewer that computational complexity and runtime measurements are important for contexting our work. We added Appendix M and Appendix N to cover these topics. In Appendix M, we break down the computational complexity for each subprocedure of an SCF step. In Appendix N, we measure the runtime of different XC functionals. We summarize the runtime in the following table.
> |Number of carbon atoms|   LDA |   SCAN |   Dick2021 |   GG-XC |   B3LYP |   $\omega$B97X |
> |---:|------:|-------:|-----------:|--------:|--------:|---------------:|
> |  2 |  0.02 |   0.02 |       0.02 |    0.05 |    0.02 |           0.02 |
> |  4 |  0.03 |   0.03 |       0.03 |    0.07 |    0.03 |           0.03 |
> |  8 |  0.06 |   0.07 |       0.08 |    0.12 |    0.07 |           0.07 |
> | 16 |  0.17 |   0.21 |       0.22 |    0.28 |    0.25 |           0.29 |
> | 32 |  0.78 |   1.03 |       1.05 |    1.15 |    1.56 |           2.09 |
> | 36 |  0.95 |   1.25 |       1.28 |    1.39 |    1.99 |           2.72 |
>
> The computational difference between EG-XC and mGGAs is relatively small compared to the runtime of hybrid functionals like B3LYP or $\omega$B97X. For mGGAs, the runtime is dominated by evaluating the density features on the quadrature grid, while for hybrid functionals, the evaluation of the exact exchange term dominates the computation. We refer to the new appendices for more details on the setup and individual runtime complexities.

---

> > ### Comment · Reviewer_Xv1K · 2024-11-25
> >
> > Thank you for your reply. It has cleared up my doubts. I am willing to increase the score.

---

### Official Review · Reviewer_NL7q · 2024-10-30

**Soundness:** 3
**Presentation:** 3
**Contribution:** 2
**Rating:** 6
**Confidence:** 3

**Summary:**

The paper introduces a new model called Global Graph Exchange Correlation (GG-XC) to improve the accuracy of exchange-correlation (XC) functionals in density functional theory (DFT). Traditional methods for approximating XC functionals struggle with capturing long-range electron interactions, leading to errors in DFT calculations. GG-XC addresses this by using graph neural networks (GNNs) to create a nuclei-centered representation of the electron density that captures long-range interactions. The approach leverages machine learning (ML) to reduce computational costs and improve data efficiency while maintaining accuracy on various energy prediction benchmarks.

**Strengths:**

1. GG-XC significantly reduces errors in DFT calculations, achieving state-of-the-art performance on energy benchmarks.
2. The method is highly data-efficient, achieving excellent results with much less training data.
3. It demonstrates robust extrapolation capabilities, performing well even on out-of-distribution and larger molecular structures.

**Weaknesses:**

1. In comparison, the force field paper prioritizes force accuracy, critical for molecular dynamics simulations, while the learned XC functional paper focuses on density accuracy, which is essential for practical DFT applications. However, the authors of the equivariant model rely only on energy metrics for evaluation, potentially leading to an incomplete comparison between the works. Despite the accessibility of force calculations via differential techniques, the equivariant model omits these, which are crucial for molecular dynamics. Additionally, density, a foundational component in DFT that governs electron distribution and influences properties such as dipole moments and band gaps, is not evaluated in the equivariant model.
2. The lack of tests on intermolecular non-covalent interactions and chemical reactions is a considerable gap. The differentiable programming approach demonstrates generalizability by successfully benchmarking against intermolecular non-covalent interaction datasets and different chemical reactions. Expanding the equivariant model's validation to include these would indicate broader reliability.
3. The lack of explicit enforcement of physical constraints may limit the accuracy in certain edge cases.
4. The model’s reliance on nuclei-based embeddings may make it less suitable for non-atomic systems.

**Questions:**

1. Given the accessibility of force calculations through differential techniques, why were these not included in the model’s experiments, could this learned energy functional be used for MD simulations or geometry optimizations?
2. Considering the fundamental role of electron density in DFT, could the model be evaluated for density accuracy and density-derived properties to ensure broader alignment with core DFT objectives?
3. Has the model been assessed for generalizability on intermolecule non-covalent interactions and diverse chemical reactions?

---

> ### Author Response · Authors · 2024-11-26
>
> We highly appreciate the reviewer's detailed feedback and are happy to hear that the reviewer agrees with our data efficiency claims. We want to address the reviewer's concerns with our revised manuscript and the following answer.
>
> **W1 + Q1 + Q2: Other DFT observables (density, forces, dipole moments, etc.)**
>
> We agree with the reviewer that these are interesting and extremely valuable aspects for evaluating XC functionals. However, we argue that improving energy accuracy is extremely valuable as it distills highly accurate quantum mechanical calculations, e.g., CCSD(T), Møller–Plesset perturbation theory, or neural-network VMC, into DFT. These methods do not necessarily permit computing other observables. To improve DFT's accuracy in such settings, this work augments the design space of ML XC functionals with efficient non-local interactions. For instance, in our new Appendix N, we find that GG-XC offers a 47% runtime reduction compared to range-separated classical hybrids like B97X while being able to learn arbitrary non-local interactions. Nonetheless, we highly encourage future work to investigate other observables.
>
> **W2 + Q3: Lack of intermolecular non-covalent interactions and chemical reactions.**
>
> While we agree that non-covalent interactions and chemical reactions are interesting applications of learnable XC functionals, such structures typically require unrestricted DFT to handle open-shell problems. We clarify this in the limitation section:
>
> > [...] to handle open-shell systems, e.g., in chemical reactions, one would need to extend the equivariant embeddings to include spin information.
>
> **W3: The lack of explicit enforcement of physical constraints.**
>
> We agree with the reviewer's sentiment that physical constraints may aid in edge cases. However, this is certainly not given due to the loose nature of the constraints. We train the mGGA from Nagai et al. [1] on 3BPA to demonstrate that constraints do not need to be beneficial. While Dick et al. [2] attempts to implement many physical constraints, no such restrictions are put on Nagai's functional. The results are summarized in the following table:
> |             |   Nagai 2020 |   Dick 2021 |
> |:------------|-------------:|-------:|
> | 300K        |         **0.91** |   0.96 |
> | 600K        |         **1.24** |   1.36 |
> | 1200K       |         **2.09** |   2.27 |
> | $\beta=120°$ |         0.85 |   **0.75** |
> | $\beta=150°$ |         0.68 |   **0.61** |
> | $\beta=180°$ |         0.66 |   **0.56** |
>
> These mixed results present no clear evidence that physical constraints yield meaningful improvements even in this out-of-distribution extrapolation setting. Other functionals like DM21 [3] come to similar conclusions. We added these results to the new Appendix O.
>
> [1] Nagai et al. "Completing density functional theory by machine learning hidden messages from molecules", 2020\
> [2] Dick et al. "Highly accurate and constrained density functional obtained with differentiable programming", 2021\
> [3] Kirkpatrick et al. "Pushing the frontiers of density functionals by solving the fractional electron problem", 2021

---

> ### Comment · Reviewer_NL7q · 2024-11-27
> **Response to authors**
>
> Thanks for the authors' reply, I would keep my score.

---

### Official Review · Reviewer_6puv · 2024-11-04

**Soundness:** 2
**Presentation:** 3
**Contribution:** 2
**Rating:** 8
**Confidence:** 3

**Summary:**

The paper introduces the Global Graph Exchange Correlation (GG-XC) functional, an innovative approach leveraging equivariant graph neural networks (GNNs) to model non-local exchange-correlation (XC) contributions in density functional theory (DFT). GG-XC addresses limitations in scalability, accuracy, and data dependency faced by previous ML-based and human-designed approximations. Through a combination of semi-local functionals and non-local density features, GG-XC captures long-range interactions and is trained using only energy targets by differentiating through a self-consistent field (SCF) solver. The empirical results demonstrate GG-XC's ability to achieve competitive accuracy with significantly reduced data requirements, even in out-of-distribution scenarios.

**Strengths:**

- Originality: GG-XC offers a novel methodology by integrating GNNs with DFT, particularly beneficial for non-local XC functional design. This approach advances ML-based DFT by capturing long-range interactions through an equivariant point cloud representation of electron density.

- Quality: The paper presents rigorous empirical evaluation across datasets (e.g., MD17, QM9, 3BPA), clearly showcasing GG-XC's data efficiency and accuracy. The use of CCSD(T) energies as a baseline adds to the robustness of the study.

- Clarity: The structural breakdown of GG-XC into nuclei-centered embeddings, equivariant message passing, non-local reweighting, and graph readout is well-organized. However, clearer explanations regarding specific design choices, such as graph readout and the integration of the SCF process, would strengthen accessibility for readers outside the immediate field.

- Significance: By bridging ML force fields and DFT, GG-XC has potential implications for more accurate and scalable quantum mechanical computations. The work promises improvements in data efficiency and applicability to larger molecules, essential for fields like materials science and drug discovery.

**Weaknesses:**

- Ambiguity in Methodological Details: Key details around the learning objectives and loss functions are unclear. For instance, it is inferred that GG-XC utilizes CCSD(T)-derived energy and density for training, but an explicit statement on this is missing. Additionally, the training schemes ˙such as backpropagating through functional derivatives) need to be explicitly written. Please clarify the optimization procedure and loss function explicitly.

- Design Choices in Non-Local Reweighting: The inclusion of the graph readout term and the reasoning behind the design choice to learn meta-GGA coefficients in GG-XC lacks sufficient justification. The authors suggest that the graph readout captures remaining non-local effects but do not clearly explain why the non-local terms in the preceding convolution is insufficient.

- Insufficient SCF Explanation: The SCF procedure, particularly the calculation of the exchange-correlation functional's differential form with respect to density, is inadequately detailed. Especially, the calculation of Fock operator is absent in the manuscript. A deeper explanation of the differentiation through the graph readout term, specifically how it integrates into the SCF iterations, is necessary.

- Computational Cost and Efficiency: The paper omits a thorough computational cost analysis. An assessment of GG-XC’s runtime compared to standard mGGA or hybrid methods would clarify the efficiency benefits. Additional experiments comparing GG-XC with conventional mGGA and hybrid methods regarding speed and computational bottlenecks would significantly improve the work. This reviewer's recommendation are the mGGA functional $\epsilon_{\text{mGGA}}$ used in the GG-XC and $\omega B97X$ functional as in the 3BPA dataset.

- Comparative Baselines: While delta-machine learning methods are included in the comparison, the paper contrasts GG-XC trained on higher-level mGGA data with models trained on lower-level LDA for delta learning. This might obscure performance differences, as delta-learning methods trained on comparable baseline data (mGGA) could offer a more direct comparison. As mentioned in the previous point, please use the mGGA functional for the baseline models.

**Questions:**

- Contribution Clarification: Could the authors explicitly clarify GG-XC’s unique methodological contributions? For readers less familiar with recent advances in this field, a succinct outline of GG-XC’s innovative aspects compared to prior methods would enhance clarity.

- Learning Objective and Loss Function: How is the GG-XC's loss function specifically defined? If CCSD(T) energy and density are indeed the targets, please confirm this assumption and clarify how functional derivatives are computed efficiently.

- Non-Local Reweighting Design Choice: What justifies the inclusion of the graph readout term when non-local terms are already introduced in the integral preceding it? Is there evidence that including the readout term substantively enhances performance?

- Differentiation Through SCF: Could the authors provide a more detailed explanation of the SCF process, particularly the derivation of the exchange-correlation functional’s differential form with respect to density? Furthermore, how is differentiation handled in the context of the graph readout term?

- Computational Efficiency Analysis: What is GG-XC’s relative computational cost compared to traditional mGGA and hybrid methods? An analysis of bottlenecks, particularly in terms of graph processing and SCF convergence, would be helpful for readers.

---

> ### Author Response · Authors · 2024-11-21
>
> We highly appreciate that the reviewer finds our work novel and would like to address the reviewer's concerns with our revised manuscript and the following answer.
>
> **W1 + Q2: Loss and gradient computation**
>
> In our revised manuscript, we added our mean squared error loss definition and a description of how we compute gradients in the experiments section. Concretely, we compute the mean squared error between the energies of the last $I_{loss}\in N_+$ SCF cycles and the target energy:
> > We follow Dick (2021) and minimize the mean squared error between the converged SCF energy and the target energy over the last $I_\text{loss}\in N_+$ steps [...]
> $$
> L = \sum_n \sum_{i=I-I_{loss}}^I (E_{n,target} - E^i_{SCF}(R_n,Z_n))^2
> $$
> [...] The parameter gradients are obtained through the total derivative of the loss with respect to the parameters of the XC functionals $\frac{dL}{d\theta}$ by backpropagating through the SCF iterations.
>
> We want to clarify that, as stated in the abstract (lines 18–19), introduction (lines 59–60), experiments (lines 349–351), and conclusion (lines 462–463), our training methodology is based solely on energy targets as these are readily available. In contrast, CCSD(T) densities are typically unavailable.
>
> **W2 + Q3: Design choices**
> We agree that ablation studies provide additional insights into our contributions. Thus, we added a new ablation study on the 3BPA dataset where we disable several components of our model. For convenience, we repeat the table here.
>
> | |300K|600K|1200K|$\beta=120$|$\beta=150$|$\beta=180$|
> |-|-|-|-|-|-|-|
> |no mGGA|7|12.89|25.85|10.99|11.16|10.82|
> |no graph readout|0.97|1.38|2.3|0.77|0.63|0.57|
> |no GNN|0.6|0.87|1.59|0.57|0.54|0.53|
> |EG-XC|**0.42**|**0.73**|**1.39**|**0.35**|**0.23**|**0.2**|
>
>
> It is apparent that each component is a significant contribution to the accuracy of GG-XC. For a detailed interpretation, we refer to the revised manuscript's experiments section.
>
> **W3 + Q4: Fock matrix definition and implementation**
>
> In our updated manuscript, we extended our SCF explanation in Appendix B to cover the construction and the computation of the Fock matrix:
> > [...] the Fock matrix is given by
> $$
> F_{ij} = H_{ij} + 2 P_{kl}J_{ijkl} + \frac{\partial E_{XC}[\rho]}{\partial P_{ij}}
> $$
> To evaluate the $E_{XC}$ and its derivative, we discretize $R^3$ with spherical integration grids around the nuclei. Specifically, for a set of quadrature points $r_i\in R^3$ and weights $w_i\in R_+$, we first compute the density and mGGA features $g(r_i)=\left[\rho(r_i), \Vert\nabla\rho(r_i)\Vert, \tau(r_i)\right]$ on these points:
> $$
>     \rho(r_i)     = \chi(r_i)^T P \chi(r_i),
> $$
> $$
>     \Vert\nabla\rho(r_i)\Vert = 2\Vert \chi(r_i)^T P \nabla\chi(r_i) \Vert_2,
> $$
> $$
>     \tau(r_i) = \frac{1}{2} \nabla\chi(r_i)^T P \nabla\chi(r_i)
> $$
> [where $\chi:R^3\to R^B$ are the basis functions.]   From this discretized input, we compute the scalar output $E_{XC}$, e.g., $E_{XC}[\rho] = \sum_{i=1}^{N} w_i \rho(r_i) \epsilon(g(r_i))$ for an mGGA, and rely on automatic differentiation via backpropagation to compute the derivative w.r.t. $P$.
>
> **W4 + Q5: Computational cost**
>
> We acknowledge that runtime and computational complexity are valuable insights. To improve our coverage of these topics, we added Appendix M, where we analyze the computational complexity of SCF and GG-XC steps, and Appendix N, where we measure runtimes for different XC functionals. To measure runtime, we implemented all functionals in JAX and measured the time of a SCF calculation. We ignore the precomputations like the core Hamiltonian, density fitting, and the evaluation of the atomic orbitals. Here, we summarize the new Figure 7 in a Table:
>
> |Number of carbon atoms|LDA|SCAN|Dick2021|GG-XC|B3LYP|$\omega$B97X|
> |:-:|-:|-:|-:|-:|-:|-:|
> |2|0.02|0.02|0.02|0.05|0.02|0.02|
> |4|0.03|0.03|0.03|0.07|0.03|0.03|
> |8|0.06|0.07|0.08|0.12|0.07|0.07|
> |16|0.17|0.21|0.22|0.28|0.25|0.29|
> |32|0.78|1.03|1.05|1.15|1.56|2.09|
> |36|0.95|1.25|1.28|1.39|1.99|2.72|
>
>
>
> Compared to hybrid functionals, our GG-XC yields a computationally efficient approach to learning non-local interactions. In contrast to hybrid functionals, GG-XC can learn arbitrary non-local interactions beyond exchange-related quantities.
>
> In computational complexity, the evaluation of mGGA-like functionals (including GG-XC) is primarily dominated by the evaluation of the density features on the quadrature grid, which requires $O(N_\text{quad}N_\text{bas}^2)$ operations. For hybrid functionals, the dominating computation is the evaluation of the exact exchange term requiring $O(N_\text{aux} N_\text{bas}^3)$ operations. For further complexity analysis, we refer the reviewer to Appendix M.

---

> > ### Author Response · Authors · 2024-11-21
> >
> > **W5: Comparative baselines**
> >
> > We would like to point the reviewer to Appendix I, where we use SCAN with the 6-31G basis set as a baseline for $\Delta$-ML. On datasets with DFT reference data, subtracting identical DFT calculations as baseline would yield no error. Thus, we performed this experiment on the MD17 dataset.
> > |Molecule|SchNet|PaiNN|NequIP|
> > |-|-|-|-|
> > |Aspirin|0.57|0.46|0.34|
> > |Benzene|0.08|0.07|0.02|
> > |Ethanol|0.26|0.17|0.08|
> > |Malonaldehyde|0.32|0.25|0.13|
> > |Toluene|0.21|0.17|0.08|
> >
> > Consistent with the reviewer's expectation, we found a significant reduction in error compared to the LDA baseline. However, it should be noted that the SCAN functional is particularly well suited for stable organic chemistry, as investigated here. Further, force fields perform very well on in-distribution data.
> >
> > Additionally, we included a new experiment where we first fit a learnable XC functional and then learned a force field on the residual error on 3BPA. This data is available in the new Appendix L. For convenience, we copy the result table here:
> > |Test set|SchNet (Dick)|PaiNN (Dick)|NequIP (Dick)|SchNet (GG-XC)|PaiNN (GG-XC)|NequIP (GG-XC)|
> > |------------------|--------|------------|--------|--------|-------------|--------|
> > |300K|0.62|0.30|0.34|0.42|**0.29**|0.30|
> > |600K|1.01|0.56|0.62|0.71|**0.52**|0.56|
> > |1200K|1.85|1.24|1.38|1.38|**1.15**|1.26|
> > |β = 120°|0.39|0.28|0.32|0.31|**0.19**|0.22|
> > |β = 150°|0.41|0.35|0.27|0.24|**0.19**|0.22|
> > |β = 180°|0.45|0.37|0.25|0.20|**0.15**|0.21|
> >
> > Here, we find GG-XC to yield consistently lower errors than Dick 2021 across all force fields, indicating that the improvements learned by EG-XC are non-identical to the force field corrections.
> >
> >
> > **Q1: Clarify contribution**
> >
> > We updated the abstract:
> > > Where previous works relied on semi-local functionals or fixed-size descriptors of the density, we compress the electron density into an SO(3)-equivariant nuclei-centered point cloud representation for efficient non-local atomic-range interactions. By applying an equivariant GNN on this point cloud, we capture molecular-range interactions in a scalable and accurate manner.
> >
> > And introduction to clarify our contributions:
> > > To this end, we propose to leverage the success of equivariant graph neural networks (GNNs) to learn non-local XC functionals through our novel Equivariant Graph Exchange Correlation (EG-XC). To enable computationally efficient non-local interactions, we propose two key innovations: (1) We compress the electron density to an SO(3)-equivariant point cloud atomic-range representation by convolving the electron density with an SO(3)-equivariant kernel at the nuclear positions. (2) We use SO(3)-equivariant GNNs on this point cloud representation to efficiently capture molecular-range information.
> >
> >
> > **Final remarks**
> >
> > We again thank the reviewer for their valuable suggestions and hope to have resolved the outstanding concerns. We are happy to engage in any further communication.

---

> > > ### Author Response · Authors · 2024-11-27
> > > **Gentle reminder**
> > >
> > > In light of the end of the original discussion period, we would again like to highlight our response. We hope that we adequately address your concerns and are interested in your thoughts. If there are any outstanding or further questions we are delighted to discuss these in the extended discussion period.

---

> ### Comment · Reviewer_6puv · 2024-11-28
>
> Thank you very much for your detailed response and for conducting additional experiments to address the questions raised in my review. As a computational chemistry researcher, albeit not specializing in DFT method development, I took long time to fully evaluate and appreciate the significance of your work. Your thorough answers and experiments have been very helpful in clarifying key points, and I would like to express my gratitude for your efforts.
> I have updated my rating to 6 in recognition of the quality and potential of your work. On the other hand, I have a few additional questions that I am curious about. These are not requests for further revisions but rather points of personal interest to deepen my understanding of your methodology.
> ### 1. Generalization to Element Types and Charged Species
> Your method seems to demonstrate strong generalization capabilities beyond the limitations of delta learning, which is very impressive. Have you tested the GG-XC functional on systems with diverse element types or charged species (e.g., ionic systems)? If so, could you share any relevant results or insights?
> I view this methodology as fundamentally different from delta learning or potential learning, targeting distinct aspects of DFT method that set it apart from those approaches.
> ### 2. Position on Jacob’s Ladder
> The cost efficiency of your method, particularly when compared to range-separated hybrid functionals, is remarkable. I am curious to know how you would position the GG-XC functional on Jacob’s Ladder of DFT accuracy.
> Specifically:
> Do you believe GG-XC can achieve accuracy comparable to or exceeding B3LYP?
> Is there a reason why comparisons with established functionals like B3LYP or ωB97X were not included in the main manuscript?
> ### 3. Analysis of TS Calculations and Optimization
> The energy landscapes shown in Figures 4, 5, and 6 are particularly striking. Based on these:
> Have you performed transition state (TS) calculations or structure optimizations using GG-XC?
> If so, have you analyzed the differences in optimized structures or energy values compared to B3LYP or ωB97X? Any results or observations on this would be very interesting.
> ### 4. System size
> How large of a system have you tested with GG-XC? For example, when calculating a molecule like $C_{38}$ with 232 electrons, how much memory does the computation require?
>
> Your work is truly inspiring, and I look forward to seeing how it impacts the computational chemistry community. Thank you again for your efforts and your time in addressing these questions!
> Best regards.

---

> ### Author Response · Authors · 2024-12-01
>
> We are excited to hear that our research has piqued your interest and are happy to discuss and expand upon your questions.
>
> **1. Inquiry: How does EG-XC perform on datasets that include more diverse nuclei and charged systems?**
>
> In principle, EG-XC is not limited to organic chemistry, but in our implementation of the differentiable SCF-solver, we have been focused on stable organic chemistry and used a spin-restricted approach to reduce the computational cost. As such, the solver is presently limited to spin-unpolarized systems.
> While evaluating charged closed-shell systems, such as salt ions, is straightforward for EG-XC, force fields are known to struggle with long-range (e.g., Coulomb interactions). We postpone such inquiries to future work, but we strongly agree that this is an interesting line of questioning, especially when focussing more on the comparison of different XC density functional approximations.
>
> **2. Inquiry: a) Where is EG-XC placed in Jacob’s Ladder?**
>
> Our method is clearly non-local (above the third rung on Jacob’s ladder [1]), but it does not fall into any of the preexisting categories to the best of our knowledge. Even so, we appreciate the sentiment of your question: Typically, both accuracy and computational cost increase as we ascend the ladder. Here, our evaluation of EG-XC indicates a promising shift, as its computational cost is only marginally higher than that of meta-GGAs, but its accuracy could surpass both that of hybrid functionals (e.g., B3LYP) and range-separated hybrids (like $\omega$B97X) which come at a significantly higher cost. Further work is required to make any definitive statements in this direction.
>
> **b) How does it compare to B3LYP and $\omega$B97X?**
>
> In general, our aim here is to develop a new functional form, rendering a comparison to general functionals (fitted to different data) difficult.
> We want to illustrate this point using the MD17 dataset on the CCSD(T) level of theory (unlike the DFT data of QM9 (B3LYP) and 3BPA ($\omega$B97X)) for the SCAN functional, as we already have the data. We note that even this physically constrained functional has free parameters fitted to reference data[2]. In the evaluation on MD17 with a fixed basis set, learnable functionals have an advantage, as their model parameters can adapt to the specific compound at hand. Additionally, the model can learn to correct for systematic errors, e.g., errors due to the finite basis set and density fitting.
>
> | MD17 (MAE $mE_h$)| SCAN | SCAN (optimal constant shift)|  Dick 2021 |  EG-XC  |
> |:------------- |-------------:|-------------:|-------------:|------------:|
> | aspirin (CCSD)| 1872 | 1.87 | 1.94 | 0.69 |
> | benzene       |  684 | 1.27 | 0.39 | 0.10 |
> | ethanol       |  216 | 1.06 | 0.85 | 0.21 |
> | malonaldehyde |  718 | 1.32 | 0.73 | 0.27 |
> | toluene       |  802 | 1.93 | 0.38 | 0.20 |
>
> As we can see, the MAE of SCAN is an order of magnitude larger than Dick 2021 and EG-XC. Even when performing an optimal constant shift (the median of the training set), which does not affect the forces, we observe an, on average, significantly worse performance compared to the learnable functionals and EG-XC in particular, despite its approximately equal computational cost.
>
> **3. Inquiry: How does EG-XC perform on structure optimization tasks and in TS calculations**
>
> We have not yet performed such tests, but we encourage the investigation of these aspects in future work.  We would be very interested in suggestions for datasets or other suitable test cases and would highly value your perspective as a computational chemist.
>
> **4. Inquiry: What is the memory requirement of an SCF calculation with EG-XC for large systems?**
>
> The largest system we have trained on is 3bpa, as shown in the main document.
> The largest system we evaluated our method on was $C_{36}H_4$, which just fitted on a GPU with 40GB memory. As you correctly point out yourself, memory (specifically GPU memory) is indeed the key limiting factor.
> During training, the caching of all intermediates tensors for backpropagation is the memory bottleneck.
> At inference, the density-fitted ERI-tensor $B$ is the memory bottleneck. For the $C_{38}$ calculation this tensor alone is 20GB.
>
> **Closing Remarks**
>
> We are encouraged by your interest in EG-XC and appreciate your insights. Your questions provide clear directions for further development and exploration, and we look forward to seeing how this work might inspire the computational chemistry community.
>
> [1] Perdew, J. P. “Jacob’s Ladder of Density Functional Approximations for the Exchange-Correlation Energy”, 2001 https://doi.org/10.1063/1.1390175.
>
> [2] Sun, J.; Ruzsinszky, A.; Perdew, J. P. “Strongly Constrained and Appropriately Normed Semi-local Density Functional”, 2015 https://doi.org/10.1103/PhysRevLett.115.036402.

---

> ### Comment · Reviewer_6puv · 2024-12-02
>
> Thank you. It's still early stage of functional learning, but the direction of the authors' research is very interesting. I will keep watching the follow-up study. I will raise the rating to 8.

---

### Official Review · Reviewer_T8Ng · 2024-11-04

**Soundness:** 3
**Presentation:** 3
**Contribution:** 3
**Rating:** 8
**Confidence:** 3

**Summary:**

This paper proposes to learn weighting coefficients for every point in 3D space to re-weight meta-GGA functional.

**Strengths:**

- The proposed method is basis-independent.
- The method shows good results on structural extrapolation and data efficiency.

**Weaknesses:**

- Since the proposed method requires computing DFT, comparing it to ML force field might not be fair due to the higher computational cost.

- The proposed ML functional still rely on the existing functional, i.e.,  meta-GGA.

**Questions:**

- The results on MD17 (Table 1) and QM9 (Figure 3) seem to be good but with mixed results. How to understand these difference in accuracy?

- How is non-local defined? Will the re-weighting also make density gradients non local?

---

> ### Author Response · Authors · 2024-11-21
>
> We highly appreciate the reviewer's feedback and hope to resolve the outstanding questions. In addition to this reply, we like to point the reviewer to the corresponding sections of the updated manuscript.
>
> **W1: Computational cost**
>
> We generally agree that the computational cost between these methods varies significantly and, thus, include $\Delta$-ML methods as a computationally more similar method in all experiments as well as the learnable XC functional from Dick 2021. To better illustrate the runtime difference, we added the new Appendix N, where we benchmark the cost of running DFT. There, we compared several DFT functionals in terms of runtime and found that GG-XC was only 7% slower than mGGAs.
>
> **W2: Reliance on mGGA**
>
> We acknowledge that the inclusion of a mGGA increases complexity but we disagree that this  is a weakness of our approach. Consistent with classical non-local functionals like B3LYP or $\omega$B97X the majority of the XC energy can be captured through such semi-local functionals. In our revised manuscript, we include an ablation study on 3BPA where we remove the mGGA. The following table summarizes the results:
> | |300K|600K|1200K|$\beta=120$|$\beta=150$|$\beta=180$|
> |-|-|-|-|-|-|-|
> |no mGGA|7|12.89|25.85|10.99|11.16|10.82|
> |EG-XC|0.42|0.73|1.39|0.35|0.23|0.2|
>
> These results support the importance of the mGGA.
>
> **Q1: The results on MD17 (Table 1) and QM9 (Figure 3) seem to be good but with mixed results. How to understand these differences in accuracy?**
>
> We agree that further interpretation aids the understanding of these results. To better communicate our findings, we elaborate further in our experimental discussion (see updated manuscript). In particular, we hypothesize that the small gap between $\Delta$-ML and XC functionals on MD17 is due to the i.i.d. nature of the dataset. In all extrapolation settings, 3BPA and QM9, we find learnable functionals yielding significant improvements.
>
> **Q2: How is non-local defined? Will the re-weighting also make density gradients non local?**
>
> We revised the introduction to yield a proper definition of non-local functionals in the introduction:
>
> > While [semi-local functionals] integrate well into existing quantum chemistry code, non-local interactions like Van der Waals forces exceed the functional class (Kaplan et al., 2023). In contrast, *non-local* functionals can capture such interactions by depending on multiple points in space simultaneously, e.g., $E_\text{XC}[\rho]=\iint_{R^3} \rho(r)\rho(r')\epsilon(g(r),g(r'))drdr'$.
>
> We extended Appendix B with a definition of the Fock matrix and how we compute the gradient of the exchange-correlation potential with respect to the density matrix. Since we differentiate through the whole XC model, the gradients of the non-local contributions are part of the XC potential $\frac{\partial E_\text{XC}[\rho]}{\partial P}$.

---

> > ### Author Response · Authors · 2024-11-27
> > **Gentle reminder**
> >
> > In light of the end of the original discussion period, we would again like to highlight our response. We hope that we adequately address your concerns and are interested in your thoughts. If there are any outstanding or further questions we are delighted to discuss these in the extended discussion period.

---

> > ### Comment · Reviewer_T8Ng · 2024-11-27
> >
> > Thank you for the clarifications and updates. I will keep my score.

---

### Meta-Review · Area_Chair_Nx3K · 2024-12-19

**Metareview:**

The submission proposes a new way of modeling and learning the exchange-correlation (XC) functional for more accurate Kohn-Sham density functional theory (DFT) calculations. The work models the electron density as atom-centered equivariant coefficients, which is processed by equivariant GNNs. The authors highlighted the effectiveness in handling long-range electron interactions, hence better accuracy and generalizability.

Reviewers have shown appreciation on the effective handling of non-local effects, the advantageous performance on benchmark datasets, efficiency over CCSD(T), and the generalization/extrapolation experiments to extend the usefulness. Reviewers also raised a few insufficiencies, including details about model design and hyperparameter choices, clear definition and ablation study of long-range effects, more evaluation metrics, and influence of lack of physical constraints as promoted in related work. In the rebuttal, the authors provided more details in the model development, detailed explanation of long-range effects and a related empirical study, and evidence that incorporating physical constraints is still debatable. These seem to have sufficiently addressed authors doubts, and two reviewers even strengthened their support.

I share the same appreciation as the reviewers on the contributions. Although both atom-centered equivariant coefficients representation of density and training a functional model with optimality enforcement are not completely novel, it is worth exploring a way that can be more suitable for a machine learning model to process, generalize, and capturing nonlocal effects. The authors also have addressed major concerns in the rebuttal. I hence recommend an accept.

Nevertheless, I still have a few points that the authors may consider for a stronger paper:
* This could be an inspiring work for machine-learning XC functionals, but at the current stage, it would be more impactful to show the promise to solve problems that conventional DFT cannot solve well, e.g., recovering macroscopic phenomenon that currently requires simulation in the CC level of theory. This may also related to the mentioned limitations, which could be reasonable for the first step, but need to be overcome for advancing the field (e.g., handling open-shell and charged systems with vast off-equilibrium structure generalizability for reactions).
* It could be a reasonable request to include more evaluation metrics as mentioned by Reviewer NL7q, considering that DFT is an electronic structure method that derives more than energy and force. The authors may also consider showing bond-breaking or angle-rotation curves to verify reasonable and useful generalization over the conformation space.
* Technically, why the radial part is not taken as the commonly chosen Gaussians, which could enjoy analytical calculation of the integrals hence getting rid of the expense and discretization error of using integral grids.
* It seems necessary to write out in the main paper how the inference stage is carried out; particularly what variable is being optimized? If it is the density integrals, then how to compute the non-interacting kinetic energy term, and how to guarantee the positiveness of the density function?

**Additional Comments On Reviewer Discussion:**

Reviewers have shown appreciation on the effective handling of non-local effects, the advantageous performance on benchmark datasets, efficiency over CCSD(T), and the generalization/extrapolation experiments to extend the usefulness. Reviewers also raised a few insufficiencies, including details about model design and hyperparameter choices, clear definition and ablation study of long-range effects, more evaluation metrics, and influence of lack of physical constraints as promoted in related work. In the rebuttal, the authors provided more details in the model development, detailed explanation of long-range effects and a related empirical study, and evidence that incorporating physical constraints is still debatable. These seem to have sufficiently addressed authors doubts, and two reviewers even strengthened their support.

I share the same appreciation as the reviewers on the contributions. Although both atom-centered equivariant coefficients representation of density and training a functional model with optimality enforcement are not completely novel, it is worth exploring a way that can be more suitable for a machine learning model to process, generalize, and capturing nonlocal effects. The authors also have addressed major concerns in the rebuttal. I hence recommend an accept.

---

### Decision · Program_Chairs · 2025-01-22

Accept (Spotlight)